# Structural insights into human brachyury DNA recognition and discovery of progressible binders for cancer therapy

Joseph A. Newman [1] ✉, Angeline E. Gavard[1,5], Nergis Imprachim [1,6], Hazel Aitkenhead[1,7], Hadley E. Sheppard[2,8], Robert te Poele [2], Paul A. Clarke [2], Mohammad Anwar Hossain [3,4], Louisa Temme[3,9], Hans J. Oh [3,4], Carrow I. Wells [3,10], Zachary W. Davis-Gilbert[3,4], Paul Workman [2] ✉, Opher Gileadi [1,11] & David H. Drewry [3,4] ✉

Brachyury is a transcription factor that plays an essential role in tumour growth of the rare bone cancer chordoma and is implicated in other solid tumours. Brachyury is minimally expressed in healthy tissues, making it a potential therapeutic target. Unfortunately, as a ligandless transcription factor, brachyury has historically been considered undruggable. To investigate direct targeting of brachyury by small molecules, we determine the structure of human brachyury both alone and in complex with DNA. The structures provide insights into DNA binding and the context of the chordoma associated G177D variant. We use crystallographic fragment screening to identify hotspots on numerous pockets on the brachyury surface. Finally, we perform follow-up chemistry on fragment hits and describe the progression of a thiazole chemical series into binders with low µM potency. Thus we show that brachyury is ligandable and provide an example of how crystallographic fragment screening may be used to target protein classes that are difficult to address using other approaches.

The T-box family of transcription factors comprises 18 members in humans and is found in a wide range of other animal species. T-box genes can function as both transcriptional activators and repressors and are required for the development of multiple cell types. Defects in these genes can result in a variety of developmental disorders[1]. They encode a highly conserved sequence-specific DNA binding domain of around 200 residues and a transcriptional regulatory domain in the C-terminus. The DNA binding domain forms an immunoglobulin-like fold and contacts DNA via an unusual insertion of an α-helix into the minor groove. All T-box proteins recognize a common consensus sequence named the T-box binding element (AGGTGTGAAA) with family member-specific variations found in the preferences for repeats, orientations (inverted or tandem), and spacing of such repeats[2]. It is still not clearly understood how these preferences relate to in vivo target gene promoter specificity as binding sites identified by Chip-Seq analysis generally show only single T-box elements[3,4] and further specificity may be achieved by interactions with other factors.

[1]Centre for Medicines Discovery, University of Oxford, Oxford, UK. [2]Centre for Cancer Drug Discovery, The Institute of Cancer Research, London, UK. [3]SGC-UNC, University of North Carolina at Chapel Hill, Chapel Hill, NC, USA. [4]UNC Lineberger Comprehensive Cancer Center, School of Medicine, University of North Carolina at Chapel Hill, Chapel Hill, NC, USA. [5]Present address: Exscientia, Oxford, UK. [6]Present address: Yusuf Hamied Department of Chemistry, University of Cambridge, Cambridge, UK. [7]Present address: Diamond Light Source Ltd, Didcot, UK. [8]Present address: Sano Genetics Ltd, Cambridge, UK. [9]Present address: Institute of Pharmacy, University of Hamburg, Hamburg, Germany. [10]Present address: GlaxoSmithKline, Collegeville, PA, USA. [11]Present address: SGC Karolinska, Centre for Molecular Medicine, Stockholm, Sweden. ✉e-mail: Joseph.newman@cmd.ox.ac.uk; paul.workman@icr.ac.uk; david.drewry@unc.edu

Brachyury (encoded by the *TBXT* gene, also formerly known as *T*) is the founding member of the family and was first identified in 1927[5]. It has subsequently been studied in detail due to its roles in the development of the notochord and posterior mesoderm[6]. Brachyury is minimally expressed after day 13 of human development except for a select few tissues (thyroid, testes, and pituitary gland), however, aberrant expression is found in various cancers. The most well-established association is with the bone cancer chordoma where brachyury is used as a diagnostic marker. Chordoma is a rare cancer (affecting around 1 in 1,000,000 people per year) that occurs along the spinal cord and is thought to originate from remnants of the embryonic notochord. Chordoma currently lacks effective targeted therapies, with primary treatment involving surgical resection followed by adjuvant radiotherapy and cytotoxic chemotherapy. Reoccurrence is common and further surgical intervention may become debilitating, with the median survival of chordoma patients being 7.7 years[7]. The *TBXT* (*brachyury*) gene is always duplicated in rare familial chordoma and in some sporadic chordomas[8] (27% of cases). It is required for growth in chordoma disease models[3,9] and has been identified by genome-scale CRISPR-Cas9 screening as the top selectively essential gene in chordoma[10]. Further associations of brachyury in chordoma are found with the observation of a common single nucleotide polymorphism (SNP) rs2305089 being strongly associated with the risk of developing chordoma in European populations[11] (odds ratio of 6.1). This variant encodes a glycine-to-aspartate substitution in the brachyury DNA binding domain. It is not clear at present what precise role this substitution plays in the protein function and in the pathogenesis of chordoma, although previous in vitro assays suggest an impact on DNA binding[12].

Further to its strong validation as an oncogenic driver of chordoma, there is evidence that brachyury is involved in various epithelial cancers where it promotes growth and induces an epithelial to mesenchymal transition and subsequently metastasis from the primary tumor site[13–15]. This is thought to be due to the presence of a T-box DNA site in the E-cadherin promoter[14] which is a key player in cell adhesion. Brachyury expression in these cancers has also been correlated with resistance to chemo- and radiotherapy[16].

Overall, these data are consistent with brachyury being the oncogenic driver in chordoma that is minimally expressed in healthy adult tissues, making it a biologically ideal therapeutic target in chordoma. However, such ligandless transcription factors have traditionally been thought to be difficult to inhibit with small molecules due to their lack of druggable pockets and polar nature of the DNA binding interface.

In this study, we address whether brachyury can be targeted with small molecules with sufficient affinity and specificity. We initially determine the crystal structures of human brachyury, and its chordoma-associated variant, bound to two different T-box binding element-containing DNA molecules. Examination and comparison of the structures provide insights into DNA recognition and allow direct comparison of WT and variant structures bound to DNA. We also develop crystal systems of both WT and variant DNA binding domains in the absence of DNA that diffract to high resolution and use these crystals in a high-throughput crystallographic fragment screen to identify ligandable pockets in brachyury. We find 29 fragments bound in 6 clusters which we use as starting points for the development of more potent binders. Here, we describe preliminary structure-guided optimization of compounds from a thiazole-containing series to a low μM level of potency which has the potential for further medicinal chemistry optimization. These compounds could lead to chemical probes of brachyury functions or be further developed into warheads for protein degradation.

## Results

### Crystal structure of WT and G177D brachyury in complex with DNA

Like many transcription factors, nearly half of the brachyury protein is intrinsically disordered (Supplementary Fig. 1) and therefore not amenable to crystallography to capture reliable structures. We therefore aimed to crystallize the DNA binding domain of human brachyury. Using the previously determined structure of brachyury from *Xenopus laevis* as a guide[17], we designed oligonucleotides containing a palindromic arrangement of T-box binding elements of varying length (22–30 nucleotides) for crystallization of human brachyury. We were able to crystallize the WT human brachyury DNA complex to 2.25 Å resolution using a construct spanning residues 41–224 with a 24-base pair DNA sequence. The chordoma-associated G177D variant brachyury was crystallized with a 26-base pair DNA sequence and diffracted to 2.15 Å resolution. Both structures feature generally high-quality electron density maps and have been restrained to standard bond lengths and angles. A summary of the data collection and refinement statistics can be found in Table 1.

Overall, the structures are very similar to previous T-box family DNA complexes, including the highly related *Xenopus* brachyury structure (92 % sequence identities and 0.9 Å RMSD). The structures feature a modified immunoglobulin-like β-sandwich fold with additional helical elements between the first and second strands and at the C-terminus (Fig. 1a). As for other T-box structures which have been obtained with near-identical DNA sequences, two copies of brachyury bind to the DNA in a 2-fold symmetrical arrangement with a small interface between subunits located towards the N-terminal end of the first β-sheet. Contacts to the DNA are made via loops between strands A and B, c and c′, and two α-helices following on from strand G at the C-terminal end of the DNA binding domain (Fig. 1b). Unusually, compared to other transcription factor families which recognize sequences via major groove interactions, the final helix is inserted into the minor groove of the DNA with two conserved aromatic side chains F213 and F217 inserting deep into the groove and making contacts with the bases. As has been noted previously[17], only two direct hydrogen bonds are made between protein and nucleobases, R69 contacting the N7 of the guanine at position 5 of the motif, and the main chain carbonyl of F213 contacting the guanine at position 7 of the motif (Fig. 1c). It has been speculated that these contacts are not sufficient to explain the observed pattern of recognition observed in in vitro site selection experiments[18]. The contact area between the two protein subunits is small (in the region of 200 Å²) and the contribution of this interface towards the possible cooperative binding on similarly spaced palindromic sites has been questioned following structures of related T-box binding proteins on the same DNA sequence which do not feature this interface[19,20]. Furthermore, the fact that palindromic arrangements of T-Box binding elements have not been identified in any known T-box target gene promoters has led to questions of what if any role cooperativity may play in vivo[3,4].

### Structural basis of T-box binding element recognition

The relatively high resolution of our DNA complex structures has allowed us to examine the DNA protein interface in detail including identification of water-mediated interactions. We have also obtained a crystal structure of WT brachyury (at 2.7 Å resolution) in complex with a single T-box binding element half-site oligo which allows us to directly compare binding interfaces and DNA distortions between single and palindromic sites. In addition to the direct contacts that define specificities at bases 3 and 5 (detailed above), recognition of a Thymine at base 4 has been previously attributed to the possibility of formation of a bifurcated hydrogen bond with the cytosine from base 5 which stabilizes the significant buckle of this base pair[19] (Fig. 1c). Hydrophobic interactions with thymine methyl groups at bases 8 and 9 to A197 and T196 likely contribute to specificities for those sites (Fig. 1c). Recognition of bases at positions 1 and 2 has been attributed to indirect mechanisms (intrinsic deformability of the DNA) as these bases are distant from the protein, which is consistent with the more relaxed specificity at these sites. The means of recognition of bases at the sixth and seventh positions has so far been more elusive,

**Table 1 | Data collection and refinement statistics for brachyury DNA complexes and ground state models**

|  | Brachyury + DNA | Brachyury D177 + DNA | Brachyury + ssDNA | Brachyury WT ground state | Brachyury D177 ground state |
|---|---|---|---|---|---|
| Space group | P 21 | P 43 2 2 | P 61 2 2 | P 41 2 2 | H 3 2 |
| Cell dimensions, a,b,c (Å) | 75.0, 37.1, 110.8 | 75.3, 75.3, 288.5 | 63.0, 63.0, 218.5 | 60.3, 60.3, 110.0 | 99.7, 99.7, 99.3 |
| Wavelength (Å) | 0.92 | 0.92 | 0.97 | 0.92 | 0.92 |
| Resolution (Å) | 29.6–2.25 (2.33–2.25) | 72.0–2.15 (2.21–2.15) | 109–2.55 (2.62–2.55) | 52.9–1.54 (1.59–1.54) | 49.8–1.42 (1.45–1.42) |
| $R_{merge}$ | 0.123 (0.689) | 0.113 (2.560) | 0.109 (2.810) | 0.092 (2.0) | 0.038 (1.198) |
| I/σI | 6.1 (1.5) | 12.9 (1.2) | 15.9 (1.0) | 12.3 (1.2) | 21 (1.2) |
| CC1/2 | 0.992 (0.653) | 0.997 (0.777) | 1.00 (0.593) | 0.998 (0.645) | 1.0 (0.562) |
| Completeness % | 97.6 (94.1) | 99.9 (98.7) | 100 (100) | 99.8 (100) | 100 (100) |
| Multiplicity | 3.4 (3.4) | 25.5 (23.1) | 17.4 (15.7) | 12.1 (10) | 8.2 (8.0) |
| No. unique reflections | 27866 (2466) | 46387 (3687) | 9158 (666) | 30913 (2746) | 35889 (2607) |
| Refinement statistics |  |  |  |  |  |
| Resolution | 29.6–2.25 | 72.0–2.15 | 109–2.55 | 52.9–1.54 | 49.8–1.42 |
| $R_{work}$/$R_{free}$ (%) | 22.56/28.82 | 22.14/25.20 | 27.33/30.18 | 22.64/24.11 | 19.75/22.31 |
| No. atoms | 4131 | 4105 | 1937 | 1462 | 1612 |
| Protein | 2843 | 2924 | 1449 | 1382 | 1425 |
| Solvent | 335 | 112 | 2 | 75 | 186 |
| DNA | 952 | 1060 | 486 | 0 | 0 |
| Average B factors (Å²) |  |  |  |  |  |
| All atoms | 38 | 69 | 82 | 36 | 25 |
| Protein | 37 | 70 | 82 | 35 | 24 |
| Solvent | 42 | 61 | 62 | 34 | 25 |
| DNA | 30 | 65 | 82 | – | – |
| Wilson B | 34 | 59 | 71 | 25.4 | 21.5 |
| R.M.S. deviations |  |  |  |  |  |
| Bond lengths (Å) | 0.007 | 0.008 | 0.006 | 0.008 | 0.012 |
| Bond angles (°) | 0.90 | 0.96 | 1.055 | 0.98 | 1.45 |
| Ramachandran plot |  |  |  |  |  |
| Favored (%) | 99 | 99 | 92 | 95 | 97 |
| Allowed (%) | 1 | 1 | 7 | 3 | 3 |
| PDB ID | 6F58 | 6F59 | 8CDN | 7HI8 | 7HI9 |

as no direct contacts are made to the protein, yet these bases are conserved in most T-box binding sites. We note that a cluster of water molecules are found in this region in conserved positions when comparing our structure with the TBX3 DNA complex obtained at 1.7 Å resolution[19]. The water network contacts the O4 of thymine at position 6, the guanine O6, and cytosine N4 at position 7. We accept that the configuration of this water network may vary depending on the DNA sequence but using known rules of hydrogen bonding donors and acceptors we can deduce that some degree of recognition is possible, for example, the presence of a H-bond donor at cytosine N7 of base 7 would specify through the water network a H-bond acceptor at O4 of base 6 (Fig. 1c).

As has been found in previous T-box family DNA structures, the DNA contains some significant distortions from regular B-form geometry, most notably a widening of the minor groove to accommodate the insertion of α-4 (Supplementary Figs. 2 and 3). These distortions are present in both the WT and G177D DNA-bound structures which have different crystal contacts and DNA-to-DNA interactions (Fig. S4b), indicating that these are not a result of the DNA environment. Both halves of the palindromic site display similar distortions, and both ends show a significant narrowing of the minor groove (from 11.7 Å in canonical B-form DNA to around 9 Å at both ends). This narrowing may be related to the presence of an A-tract (defined as a stretch of 4 or more A-T base pairs without a TpA step) on either end of the palindromic sequence, which is known to facilitate DNA bending and narrowing of the minor groove[21]. Given that only the last two T-A bases of the A-tract are within the T-box motif it is not clear if this narrowing is a feature in T-box family DNA recognition. Comparing the structures of brachyury bound to a palindromic DNA with a 12-base pair single site (PDB 8CDN and Table 1, column 3) reveals that most of the contacts at the interface are maintained (Supplementary Fig. 3) although the end narrowing of the minor groove is not present in the single site DNA which also does not contain a full A-tract sequence (Supplementary Fig. 3), suggesting perhaps that this may just be a feature of the palindrome used for crystallization.

## Comparison of WT and G177D structures

As would be expected with only a single substitution, the structures of the WT and G177D variant are very similar (RMSD 0.8 Å) with only minor differences seen in the ordering and conformations of various loops that are presumably flexible (Fig. 2a). The G177D substitution itself lies in a loop between strands F and G and is situated away from the DNA interface on the opposite side of the protein. The conformation of this loop is significantly altered (Fig. 2a) with a new salt bridge formed between the substituted D177 residue and R174, possibly explaining the altered conformation in the variant structure. Furthermore, the G177D substitution is within a GGP peptide sequence that lies in a restricted region of Ramachandran space. The PHI and PSI angles adopted by the glycine are not permitted for a non-glycine residue, particularly in the case of a pre-proline which is more restrictive due to steric hindrance around the proline Cδ. The crystallographic B factors indicate this loop is fairly mobile in both the WT

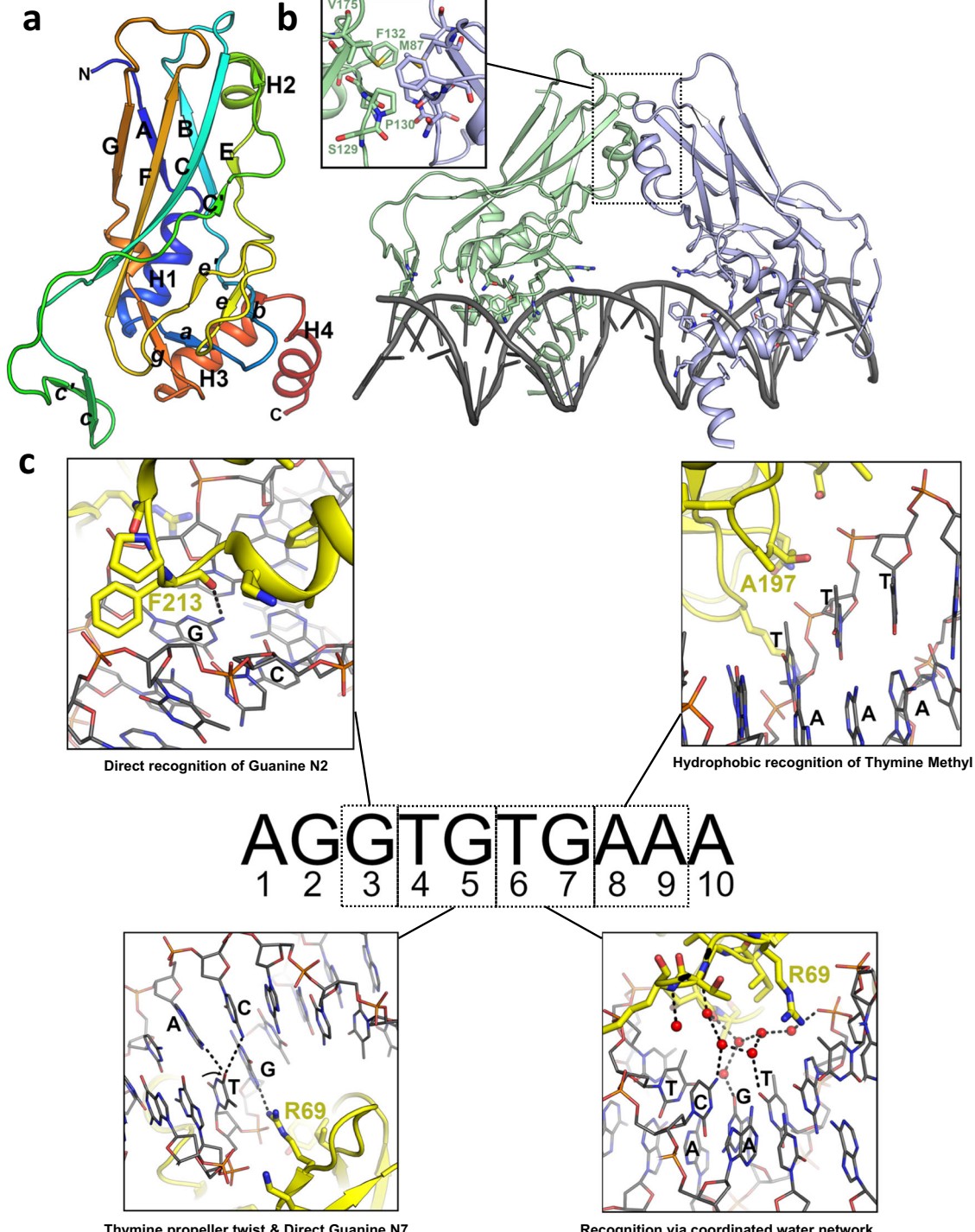

**Fig. 1 | Structure of brachyury in complex with DNA. a** Overall structure of human brachyury DNA binding domain with secondary structure elements labeled in accordance with the immunoglobulin fold nomenclature. **b** Structure of human brachyury bound to a palindromic DNA containing an inverted repeat of the T-box recognition element. Two brachyury protomers bind to each DNA half site with an unusual insertion of a helix into the minor groove and form a small interface between the two chains which is shown in the inset. **c** Details of the recognition of specific DNA base pairs by brachyury. For each nucleotide position in the T-box recognition element, the mode of recognition is indicated in the inset box with key residues and nucleotides labeled.

and G177D brachyury structures with B factors slightly higher than neighboring residues for both structures, although the potential stabilizing influence of crystal contacts cannot be ruled out (Supplementary Fig. 4a, b). Whilst the differences in this loop may be significant on a local level, we do not see a possible way for these changes to be transmitted through the protein to the DNA binding site. The G177D substitution is however close to the site of the small protein

interface that is created between subunits when binding on DNA containing T-box binding sites in a palindromic arrangement. Thus, it is plausible that the substitution may affect the cooperativity of DNA binding as has been suggested previously[12]. Analysis of the interfaces between subunits using the program PISA[22] reveals some very minor differences in the interface areas (210 Å² for the G177D versus 219 Å² for the WT) and in the calculated energetic contributions toward toward

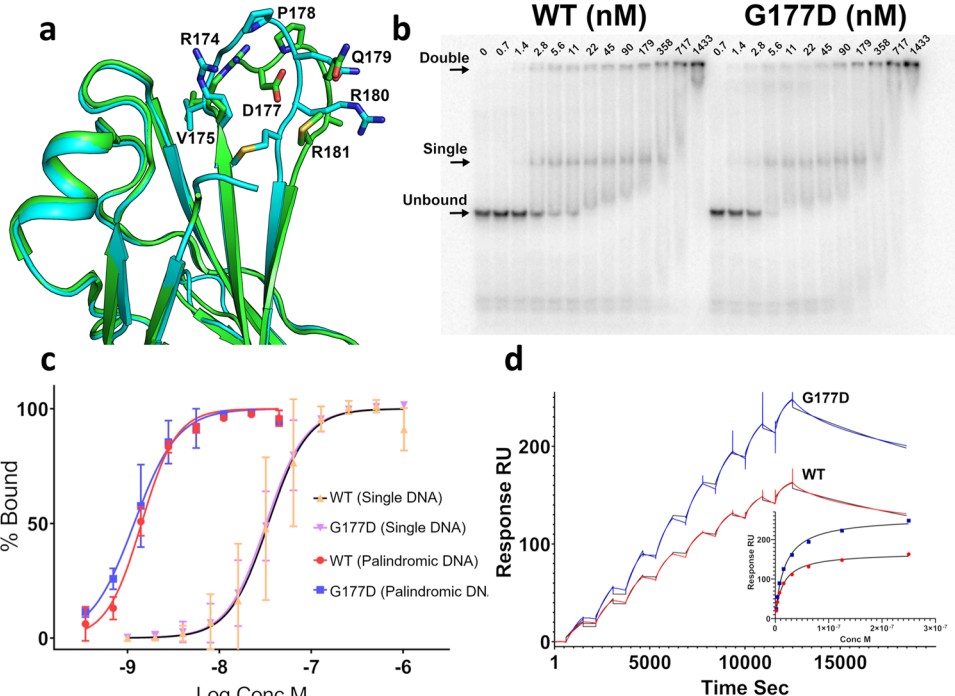

**Fig. 2 | Analysis of the G177D variant on brachyury structure and DNA binding.**
**a** Comparison of the structures of WT (Cyan) and G177D (Green) DNA binding domains in the vicinity of the loop 175-181. **b** Representative electrophoretic mobility shift assay (EMSA) gel of WT and G177D full-length brachyury binding to a 50 bp DNA with palindromic repeat of the T-box recognition element.
**c** Quantification of EMSA data comparing WT and G177D brachyury binding to a single site (FGF8 promoter sequence) and palindromic repeat of the T-box recognition element. Error bars are shown as means ± the standard deviation from

analysis of three independent experiments. Source data are provided as a Source Data file. **d** Analysis and comparison of WT and G177D brachyury binding to palindromic DNA by surface plasmon resonance. The main plot shows responses generated for sequential injections of DNA with full-length WT or G177D brachyury immobilized on the sensor surface. The black line shows the fit to the data using a bivalent analyte model and inset shows the same data fitted using a concentration-response curve.

---

complex formation (−4.8 Kcal/mol versus −4.5 Kcal/Mol); we consider these differences to be unlikely to have a role in the biological activity of brachyury.

### The G177D variant does not substantially change DNA binding affinity

Although the structural differences may be small, it is plausible that they contribute towards a difference in the cooperative binding at sites with inverted repeats of the T-box binding element. We tested this using in vitro DNA binding assays and compared the full-length WT and the G177D variant by electrophoretic mobility shift assay (EMSA) on a range of DNA probes containing palindromic, single sites, and natural promoter sequences. We found as others have observed for various T-box family members, a preference for palindromic repeats with an apparent $K_d$ of around 1 nM (Fig. 2b, c). Binding of a single site probe or a probe from a natural brachyury target promoter (sequence identified from Fibroblast Growth Factor FGF8) gave a lower apparent $K_d$ of 30–40 nM (Fig. 2c and Supplementary Fig. 5a). Two shifted species could be observed on the gel. For the palindromic probe, the upper band is of low mobility and migrates only minimally into the gel. We assume these two bands represent singly and doubly bound species; this assumption is consistent with the expected mobilities and mass-to-charge ratios of the complexes, and experiments using the brachyury DNA binding domain alone (residues 41–224) which show the same banding pattern but both upper and lower bands are able to migrate into the gel distinct from any aggregated proteins that may be present near the wells (Supplementary Fig. 5). From examination of the bands on the EMSA assay there does appear to be some degree of cooperativity as both upper and lower bands appear at approximately the same point in the titration, rather than the upper band lagging

behind the lower for a noncooperative independent binding scenario. We do not see any significant differences in overall binding affinity when comparing the WT and variant proteins, with the dissociation constants [1.4 nM WT (95%CI 1.32–1.53) and 1.2 nM G177D (95%CI 1.03–1.35)] from the quantification of the data being not significantly different if using a 95% confidence interval (Fig. 2c). On the other hand, differences in the degree of cooperativity are less certain. The upper and lower shifted bands on our gel appear to follow a broadly similar concentration-response pattern and the Hill slopes from the overall quantification [2.3 WT (95%CI 1.93–2.74) and 1.8 G177D (95%CI 1.46–2.35)] are similar. We note that these bands were not quantified independently due to uncertainties over the exact composition. Also, the DNA probe in our experiments is present at a similar concentration range to the apparent dissociation constants, giving the possibility for ligand depletion effects to affect the apparent dissociation constants, although this effect is equal across WT and G177D variants.

We have also tested the binding of a palindromic DNA sequence to brachyury WT and G177D via surface plasmon resonance (SPR) with the protein immobilized onto a streptavidin sensor surface. A clear high-affinity binding interaction can be observed which can be fit as a concentration response with apparent dissociation constants of 14.8 ± 2.4 nM for the WT and 17.9 ± 2.9 nM for the G177D variant. Fitting the data using a kinetic model is only possible with a bivalent binding model consistent with two binding sites on the DNA (Fig. 2d and Supplementary Table 1). The reason for the cooperative binding has been the subject of some debate within the T-box family, as the small interface formed between brachyury subunits is not present in other T-box family member DNA structures[19,20] although these family members do show some preference for palindromic DNA in in vitro assays[23]. We suggest a possible explanation for this phenomenon, that the

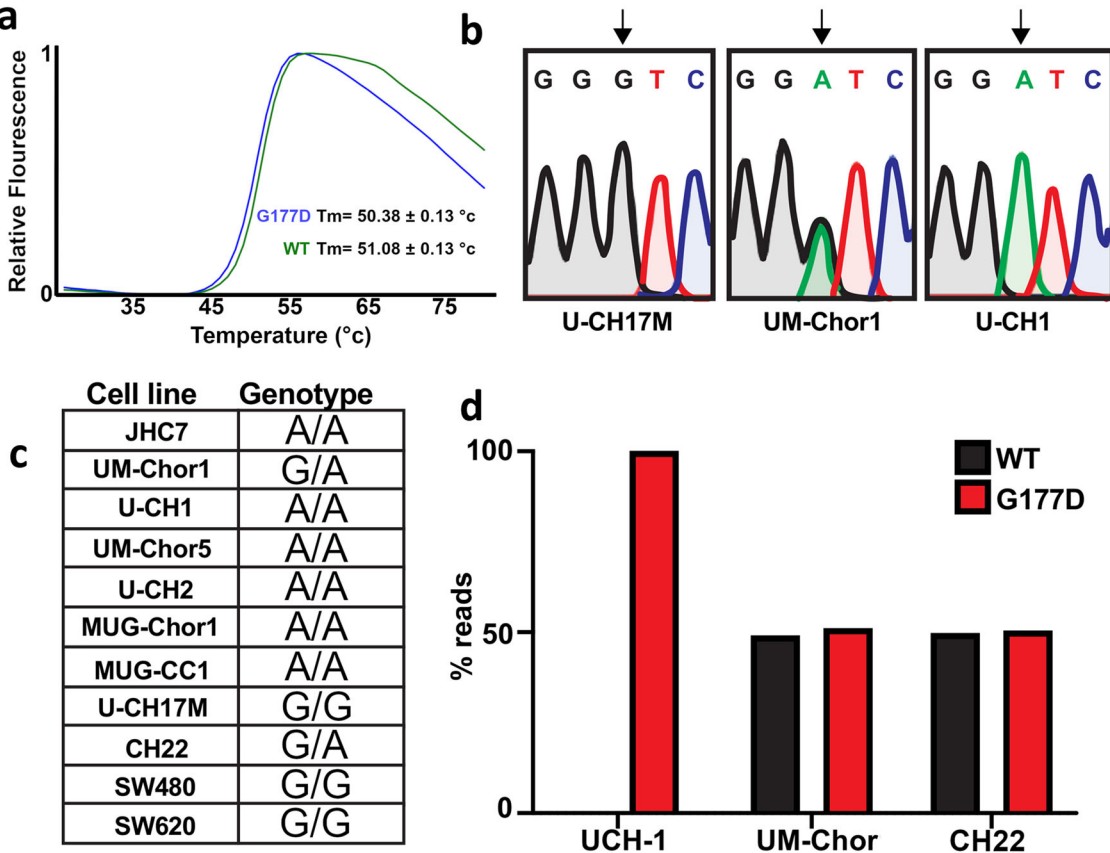

**Fig. 3 | Analysis of the common G177D variant thermal stability and prevalence in human chordoma cell lines. a** Differential scanning fluorimetry (DSF) measurement of the thermal stability of WT and G177D full-length brachyury proteins. Source data are provided as a Source Data file. **b** DNA sequencing chromatograms of three chordoma cell lines in the vicinity of the polymorphism. WT brachyury is encoded by a Guanine nucleotide at the 3rd position whilst G177D variant has an Adenine. **c** Table of the genotypes of various chordoma cell lines at the SNP rs2305089 (G signifies WT and A the G177D variant). **d** RNA-Seq analysis of chordoma cell lines showing approximately equal transcription of WT and G177D alleles in the two heterozygous human chordoma cell lines UM-Chor1 and CH22.

widening of the minor groove for binding at one site lowers the energy barrier (due to proximity) for the widening of the minor groove for the nearby binding of a second protomer. The fact that the two T-box sites are arranged in the inverted orientation and in the closest possible proximity for binding both sites without steric clashes would fit with this 'through DNA' model. The small interface between subunits may play an additional role in a subset of T-box family members such as brachyury.

## The chordoma-associated variant is transcribed equally to WT in cell lines

While the G177D variant does not substantially change DNA binding in vitro, we aimed to further characterize its biological relevance. Previous studies investigated the role of the G177D variant using engineered isogenic cellular systems that expressed only WT or G177D brachyury[3]. Each of these isogenic cell lines was viable, suggesting that the G177D brachyury variant is not a sole chordoma driver. Furthermore, there was no significant difference in the downstream brachyury target genes as identified by ChIP-sequencing[3]. We aimed to understand the consequence of the G177D variant within its endogenous context as other reasons for its association with chordoma could lie in protein stability or expression levels. However, examining the stability of WT and G177D brachyury by differential scanning fluorimetry (DSF) shows only a small melting temperature TM shift of -0.7°, with the WT variant appearing to be very slightly more thermostable than the chordoma-associated variant (Fig. 3a) and thus is likely not an explanation for the association of the G177D SNP with chordoma, although we cannot rule out differences in transcriptional efficiency or

metabolic stability owing to interactions with cellular degradation machinery. To characterize the prevalence of the G177D variant in chordoma cells, we obtained a panel of 9 chordoma lines and genotyped them for the presence of the G177D variant. These cells included lines derived from both sacral and clival chordomas, primary and metastatic tumor sites, and the pediatric chordoma cell line UM-Chor5. Sanger sequencing of exon 4 of the *TBXT* locus confirmed the presence of the WT (G) or G177D (A) variant (Fig. 3b, c). We find that the majority of chordoma cell lines (6/9) genotyped are homozygous for the chordoma-associated SNP. The majority of homozygous variants across the chordoma lines mirrors previous patient findings[11]. Interestingly, the chordoma cell line U-CH17M, derived from a metastatic chordoma tumor[24], does not encode the G177D variant and is the first chordoma cell line without the variant identified to date. SW480/ SW620 colorectal cancer cells, control non-chordoma cell lines which have been shown to express brachyury[9], also do not encode the G177D brachyury variant. Given a minority of chordoma cell lines are also heterozygous for the brachyury variant, we aimed to determine the expression levels of the WT vs G177D alleles. Using publicly available RNA-sequencing data[10], we confirm that each variant in the heterozygous cell line UM-Chor1 is equally transcribed. Together, these data, coupled with previous findings[3], suggest a brachyury-directed therapeutic will most likely need to target both the WT and G177D variant brachyury.

## Crystallographic fragment screening of brachyury

The causal role of brachyury in human chordoma cancers coupled with the absence of detectable expression of brachyury in most adult

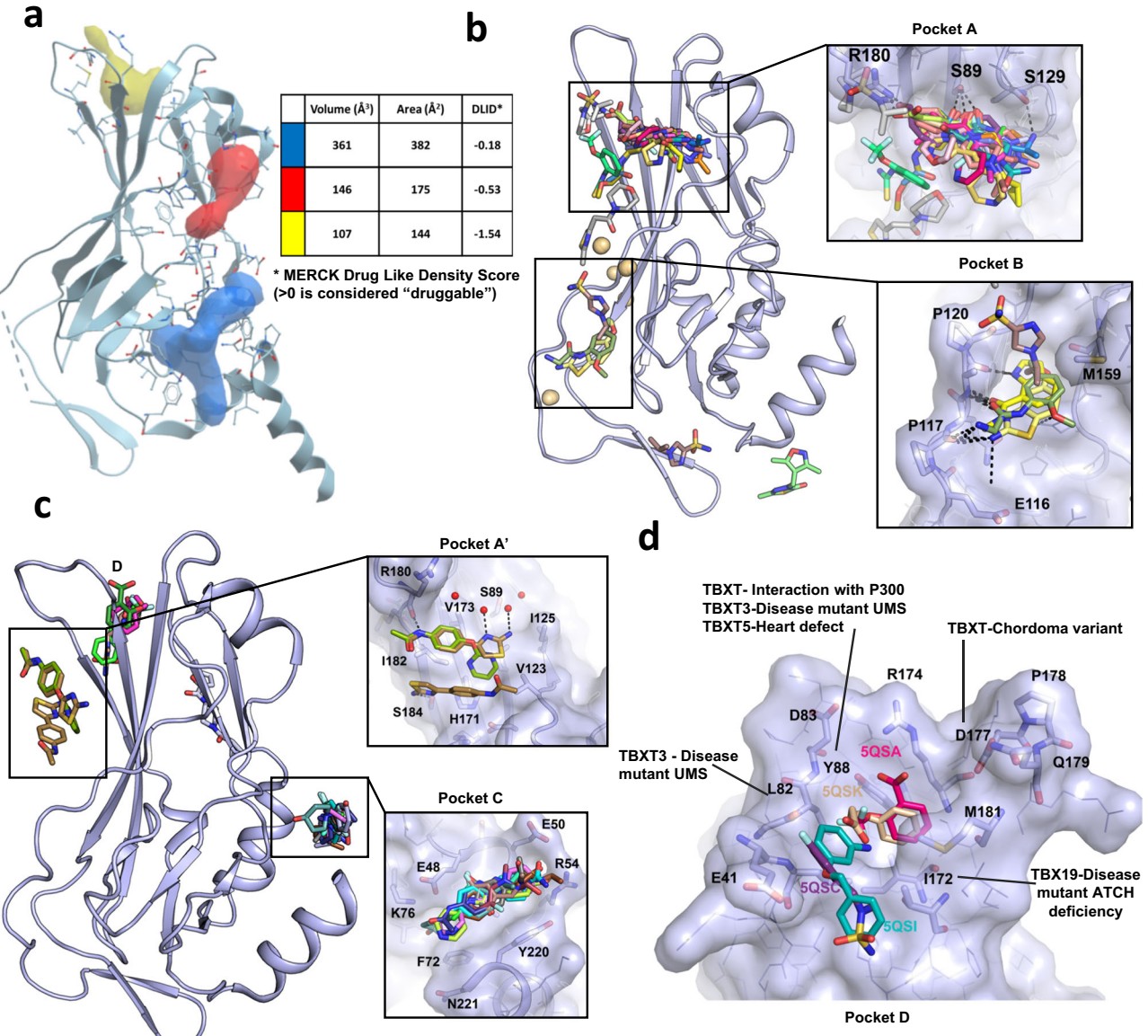

| Volume (Å³) | Area (Å²) | DLID* |
|---|---|---|
| 361 | 382 | -0.18 |
| 146 | 175 | -0.53 |
| 107 | 144 | -1.54 |

\* MERCK Drug Like Density Score
(>0 is considered "druggable")

**Fig. 4 | Druggability analysis and fragment screening of brachyury. a** Pocket analysis of brachyury DNA binding domain. Pockets are depicted as solid blobs of red, blue, and yellow with volumes, areas, and predicted druggability scores shown in the inset table. Druggability scores are calculated using ICM-Pro according to the method of Sheridan et al.[28]. **b** Fragment screening results for WT brachyury DNA binding domain with hotspot pockets shown in the insets in a surface representation with key residues labeled. **c** Fragment screening results for G177D brachyury DNA binding domain with hotspot pockets shown in the insets in a surface representation with key residues labeled. **d** Close-up view of the fragments bound to pocket D with residues involved in disease mutations in other T-box family members labeled.

human tissues, indicate that, from the point of view of potential effectiveness and therapeutic index, it is an ideal drug target biologically for drug discovery. However, ligandless transcription factors have traditionally been considered very difficult to target due to the presence of intrinsic disorder and lack of well-defined pockets for small molecule binding[25]. We have tested this premise and performed a druggability analysis of our brachyury structures using the ICM pocket finder algorithm (Fig. 4a)[26,27]. As expected, only three pockets of moderate volume could be found that were not predicted to be druggable based on the Drug-Like-Density metric[28]. Similar results were obtained using the ligandability analysis in the canSAR knowledgebase[29], with the largest pocket displaying properties on the lower limit for a classically a druggable site (kinase) and similar to a challenging yet druggable target (Bcl 2) (Supplementary Fig. 6). Such calculations can serve as useful barometers, but new inhibition

modalities such as proteolysis targeting chimeras (PROTACS) offer promise even for these difficult targets[30,31].

To further investigate the potential for direct inhibition or degradation of brachyury, and to discover starting points for structure-based drug discovery, we have performed a crystallographic fragment screen on both WT and G177D variant brachyury DNA binding domains. Crystallographic fragment screening relies on direct detection of fragments in electron density maps, the reliability of which is greatly enhanced with crystals that diffract to high resolution. For X-ray fragment screening in the absence of DNA, we used a shorter construct (amino acids 41–211) lacking the C-terminal helix that inserts into the DNA minor groove, which was observed to be flexible in our early DNA-free crystallization efforts. WT and G177D DNA-free crystals were obtained in different conditions with different crystal forms although both generally diffracted to around 1.6 Å. The structures of

WT and G177D brachyury in the absence of DNA are almost identical to their counterparts bound to DNA (R.M.S.D. 0.36 and 0.37, respectively), including the significant differences observed around the G177D variant despite different crystal packing interactions, thus validating that these differences are not crystallographic artifacts (Supplementary Fig. 4). A total of 609 fragments were soaked into WT crystals and diffraction data were collected and analyzed using the PanDDA algorithm[32], yielding a total of 30 fragment hits across 27 datasets. For the G177D crystals, a total of 616 fragments were soaked yielding 17 fragment hits across 16 datasets (Supplementary Figs. 7 and 8, and Supplementary Table 2). In both fragment screens hits were identified ranging from high occupancy ligands with good quality electron density, to low occupancy ligands with only weak density if analyzed using a conventional $2F_o$-$1F_c$ electron density map (Supplementary Table 2). For these lower occupancy ligands, the main evidence for ligand binding comes from the PanDDA event maps which are shown in Supplementary Figs. 7 and 8. Comparisons between the two datasets show (perhaps surprisingly) very little overlap between the fragment hits, despite the same fragment library being soaked. Presumably, this is due to a combination of the different accessibility of sites in the two crystal forms (Supplementary Fig. 9) and the sensitivity of X-ray fragment binding to minor changes in conformation or chemical conditions.

In the WT crystals a significant hotspot (pocket A) was identified near the N-terminal end of strand *c*, with 24 fragments bound, the majority of which make a hydrogen bond to S89 with additional contacts to R180 and S129 (Fig. 4b). A further 4 fragments are bound to a pocket (pocket B) formed between strands *c'* and *e'* which make polar contacts to the main chain of loop 116–120 (which was found to be partially disordered in the DNA complex structures), and side chains of M159 and E116 (Fig. 4b). The WT fragment screen was performed in crystallization conditions containing cadmium chloride with typically 5 cadmium ions are bound to surface sites in the protein making contacts to exposed cysteine and histidine residues. We acknowledge that the presence of these ions may have some influence on fragment binding although only a single fragment was observed to bind primarily to a cadmium ion. In the G177D form, a hotspot pocket (pocket C) was located near the C-terminal end of the final helix which contains 9 fragments that make polar contacts to R54, E48, and K76 (Fig. 4c). Although this pocket lies on a crystallographic 2-fold symmetry axis, the fragments are close to the DNA interface (~8 Å away) and the pocket is significantly larger and extends down to the DNA interface in the DNA complex structures with the longer construct. Three fragments were observed to bind in a roughly equivalent site to pocket A in the WT (pocket A') although these fragments engage in more hydrophobic-type interactions with I182, L91, and V123 (Fig. 4c). Finally, 4 fragments bound to a pocket near the N-terminus (pocket D) which appears to be induced partially by ligand binding. Two fragments with a benzene ring occupy a relatively buried cavity that was also observed to bind a 2-methyl-2,4-pentanediol (MPD) molecule (a component of the crystallization solution) in the G177D DNA complex structure, and make contacts to R174, Y88, and M181. This pocket is also lined by G/D177 and binding may be specific to the G177D variant conformation (Fig. 4d). Across both screens only two fragments (PDB entries 5QRM and 5QRW) are bound to the regions containing the DNA binding interface, in line with the general view in drug discovery that such interfaces, being relatively flat and polar, do not generally contain tractable pockets to support the binding of small molecules.

It is not known whether any of the pockets identified are sites that upon compound binding will lead to inhibition of brachyury or modulation of its function, although pockets B and C approach to within around 8 Å of the DNA interface with the potential for fragment growth in that direction. Pockets A and A' are close (within 3 Å) to the potential dimer interface that is formed when binding to palindrome T-box sites and may be extended to disrupt DNA cooperativity without directly having to compete with DNA binding. Finally, pocket D is distant from both dimerization and DNA binding sites but has been implicated to have a likely role in downstream signaling due to the observation of dependence for a tyrosine residue at position 88 for interaction with P300, a component of the histone modification machinery[33]. Residues lining this pocket are well conserved in the T-box family and have been shown to be important in vivo as clusters of point mutations from several different human genetic diseases map to this site (Fig. 4d), and mutational analysis in the murine T-bet protein indicates critical roles for this pocket in interactions with permissive chromatin re-modelers including KDM6A and KDM6B[34]. Given the importance of this pocket in other T-Box family members, it is tempting to speculate that it plays a role in brachyury for the interaction with its own downstream effectors, possibly explaining the G177D variant association in chordoma.

## Structure-guided optimization of fragments to potent brachyury binders

Encouraged by the discovery of ligandable pockets on brachyury, we have initiated a medicinal chemistry campaign to optimize the potency of fragment-derived molecules. We have used biophysical binding by SPR as our primary assay to measure the potency of compounds irrespective of the potential to inhibit brachyury activity. Only compounds that displayed high-quality sensograms with concentration-dependent responses were included for analysis (Supplementary Table 2). This biophysical approach also enables the discovery of potent binders that, even if they are not inhibitors of brachyury function, could be used as warheads to induce the degradation of brachyury through a PROTAC modality. This degradation approach has been validated for brachyury in chordoma cells using the inducible degradation dTAG system[3].

One of the promising chemical series is based on a thiazole fragment found in pocket A' (PDB: 5QS9) which bound with two molecules near a solvent-exposed hydrophobic surface patch that is formed by the second β-sheet containing strands *c,c',f* and *g* (Fig. 4c). One molecule of thiazole **1a** shows binding to R180 and two exposed solvent molecules, and the other molecule of thiazole **1a** shows binding to S184 (Fig. 5a). Initial work aimed to induce a selectivity bias between the two different binding modes. Chemistry methods and compound characterization are provided in the Supplementary Information. Replacement of the 2-NH$_2$ of the thiazole (Table 2, Entry **1a**) with a 2-morpholino moiety led to a compound (Table 2, Entry **1b**) with measurable binding affinity on SPR that could be soaked into our crystals, revealing a single mode of binding equivalent to the R180-interacting pose of the original thiazole hit (Fig. 5c). This fragment retains the hydrogen bond to R180 with most of the rest of the interaction mediated by hydrophobic interactions with nearby side chains of residues L91, V123, I125, V173, and I182 which form an unusual cluster of surface-exposed hydrophobic residues that are conserved amongst other T-box family structures (Supplementary Fig. 10). Further improvement was obtained by conversion of the acetamide to a cyclopropylacetamide which provided the most notable improvement in this limited series of analogs (Table 2, Entry **2** and Supplementary Table 3) increasing the apparent binding affinity to the low μM range (14–20 μM) as measured by SPR (Table S3). The cyclopropyl group has the potential to make favorable hydrophobic interactions with the side chains of residues I182 and M181. We hypothesize that steric clashes with crystallographic neighbors thwarted attempts to generate structures with ligands containing these cyclopropyl groups. Additional modifications at this position were also tolerated, including replacement with an ortho-fluoro methylsulfone (Table 2, Entry **6**). The morpholine group could be replaced with piperazin-2-one or 3,5-*cis*-dimethylmorpholine (Table 2, compounds **3** and **4**). We were able to employ either 2,4-substituted or 4,6-substituted pyrimidines (Table 2, Entry **7**–**10**) as isosteric replacements of the thiazole ring, and compounds with these modifications retained binding activity

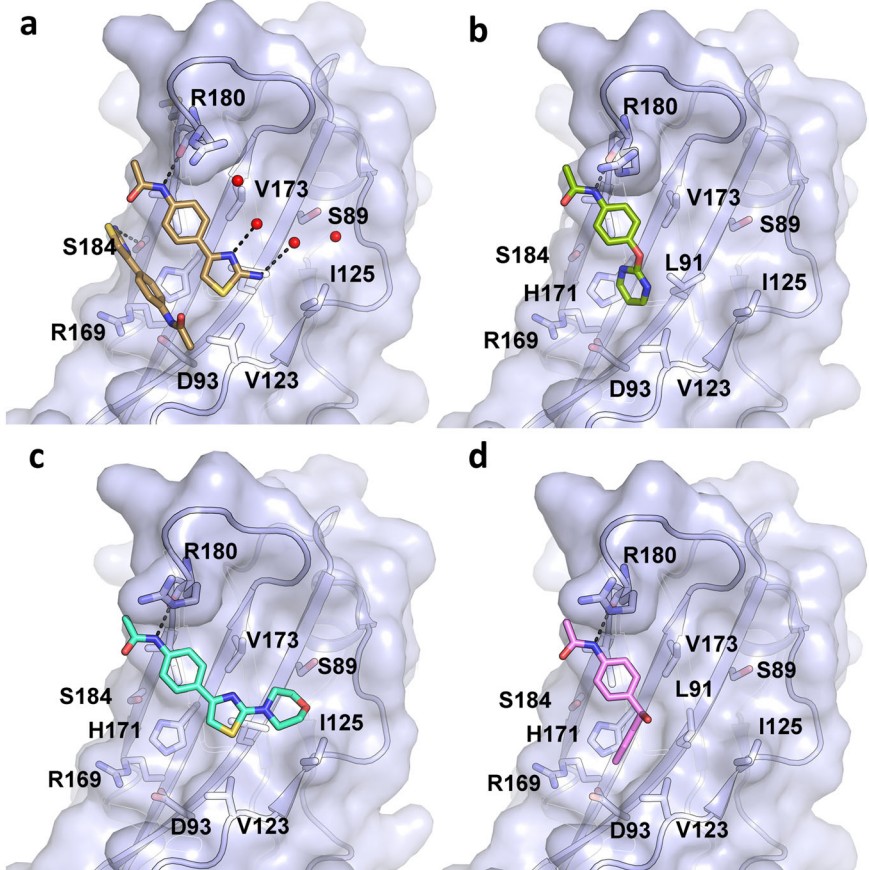

**Fig. 5 | Structures of fragment derived hits binding to pocket A'. a** Crystal Structure of 5QS9 in pocket A' of brachyury. **b** Crystal Structure of 5QSD in pocket A' of brachyury. These two compounds occupy very similar binding modes in this space. The primary interaction in these fragments is a hydrogen bond to ARG180 from the amide functional group (this can be observed in both 5QS9 and 5QSD).

**c** Crystal Structure of Compound 1B (PDBid 7ZK2) in pocket A' of brachyury. Showing contact to ARG180, and further interactions with the hydrophobic pocket surface. **d** Crystal structure of compound **11** in complex with brachyury (PDBid 8A7N) forming hydrophobic contacts with a conserved hydrophobic cleft.

(Supplementary Table 2). Moving from the 5-membered thiazole to 6-membered pyrimidines provides an additional vector for further optimization of the series and an additional position for exploration of exit vectors for bivalent degraders (for example, compound **9**). In our original pyrimidine fragment (PDB 5QSD, Fig. 5b) the pyrimidine moiety inserts into a conserved hydrophobic cleft close to L91, V123, and a polar triad of residues (H171, R169, and D93) that form a network of hydrogen bonds (Figs. 5b and S6). In our initial fragment screen, we also found a similar N-phenylacetamide in the A' pocket which contained a diphenylmethanol group (Table 2, Entry **11**, and Fig. 5d). Further exploration of the secondary alcohol with chlorobenzamide (Table 2, Entry **12**) or methoxybenzamide (Table 2, Entry **13**) groups provided significant increases in potency for this limited series of compounds as measured by SPR (Supplementary Table 3).

## Discussion

Directly targeting ligandless transcription factors as a cancer therapy has huge potential[35], however, with a few notable exceptions[36–38] this potential has yet to be realized. Direct targeting of transcription factors with small molecules remains challenging due to the combination of frequent intrinsic disorder, lack of defined druggable pockets, and the dynamic nature of transcription factor complexes. In this study, we have investigated the potential for direct targeting of the human brachyury transcription factor, which has a causal role in chordoma cancers, using a structure-guided approach. We have determined structures of human brachyury both alone and in complex with DNA and have used our high-resolution APO form crystals to perform

crystallographic fragment screening. This screen identified several ligandable pockets on the surface of brachyury, and we have been able through structure-guided chemical optimization to progress fragment hits to binders with apparent dissociation constants in the low μM range. Very little overlap was observed in the fragment hits between the screens performed with the WT and the chordoma-associated G177D variant. We attribute this to the fact that the two screens were performed on different crystal forms and the influence of crystal packing and crystallization conditions, rather than the structural differences between the two forms, the exception being some of the hits in pocket D which do contact variant residues.

It is important to note that these compounds have been validated as binders and are not yet inhibitors of brachyury function. As suggested by our RNA-Seq analysis an effective therapeutic targeting brachyury may potentially need to target both the WT and G177D variants, and we expect this to be the case for our optimized compounds. However, to properly address the question of the relative protein levels of WT vs G177D brachyury in chordoma, and hence the need for a therapeutic to target both forms, would require a large-scale mass spec-based proteomics approach using multiple cell lines and human tissues. Structural data generated to date on this series show that they bind a considerable distance from the variant site and DNA interfaces. We have tested compounds from this series using an in vitro fluorescence polarization-based DNA binding assay and do not find any evidence for inhibition of DNA binding, although we note that it is possible there are functional consequences for binding at this site in the cell. Our compounds may serve as starting points for further

**Table 2 | Chemical structures and binding affinity of compounds measured by SPR**

| Compound (PDB code) | Structure | Kd µM | Compound | Structure | Kd µM |
|---|---|---|---|---|---|
| 1a (5QS9) | | – | 7 | | 14 ± 9 |
| 1b (7ZK2) | | 316 ± 90 | 8 | | 16 ± 7 |
| 2 | | 19 ± 14 | 9 | | 30 ± 9 |
| 3 | | 14 ± 9 | 10 | | 8 ± 3 |
| 4 | | 21 ± 10 | 11 (7ZKF) | | 340 ± 8 |
| 5 | | 84 ± 9 | 12 | | 49 ± 6 |
| 6 | | 25 ± 7 | 13 | | 17 ± 9 |

PDB codes are given in parenthesis for compounds that were successfully soaked into brachyury crystals. Errors are shown as ±the standard error based on the affinity fit to the concentration-response curves.

development into PROTAC warheads to induce the degradation of brachyury. Future PROTACS potentially derived from our lead compounds and binding to either or both forms, could be used as chemical probes to determine the relative functional involvement of the WT and G177D variant in models of chordoma—and hence inform on the best therapeutic strategy.

The compounds we have identified bind to relatively shallow surface pockets, one of which has an unusual lipophilic character due to several surface-exposed hydrophobic residues. Importantly these pockets were not identified as easily druggable by a computational analysis and hits binding to this pocket would be unlikely to be detected by conventional high-throughput screening if using a DNA binding readout. Our results demonstrate the power of the structure-guided fragment-based approach for identifying binders for classes of proteins that are otherwise generally considered intractable for drug discovery. Whilst low µM binders are not sufficiently potent for pharmacological inhibition in a classical occupancy-driven model, such moderate affinity binders have previously been shown with other protein targets to be effective as PROTAC warheads[39]. As PROTAC warheads they participate in what has been called event-driven pharmacology, and are not required to occupy the binding site for

extended periods of time—and indeed may exert their effects through transient interactions that allow multiple rounds of degradation[40,41]. The work we have described here adds to the notion that emerging technologies, including fragment screening by X-ray crystallography, will soon move transcription factors out of the undruggable category, and a new reality will emerge in which transcription factors are valuable and tractable drug discovery targets[42].

## Methods
### Cloning expression and purification
Constructs for WT and G177D brachyury DNA binding domain (residues 41–224) (used for DNA complex crystal structures) in pET28 were obtained as a gift from Michael Miley, University of North Carolina. Constructs for the WT and G177D truncated DNA binding domain (residues 41–211) (used for fragment screening crystals) were cloned into pSUMO-LIC vector using ligation-independent cloning to produce proteins with an N-terminal His-SUMO tag. Full-length WT and G177D brachyury (used in the DSF and SPR assays) were cloned into the pHGT-Bio vector by ligation-independent cloning for expression of brachyury with a N-terminal 9His-GST tag and C-terminal AVI tag for biotinylation of the target protein by co-expression with BirA enzyme

in the presence of biotin[43]. All plasmids generated in this study have been deposited in Addgene.

From glycerol stocks, bacteria (*Escherichia coli* Rosetta DE3 strain) were inoculated in 15 ml of 1 × TB in a 50 ml tube with kanamycin 0.05 mg/ml and 0.034 mg/ml of chloramphenicol and grown overnight in a shaker at 37 °C, 250 rpm. The following day, 4 ml of the overnight culture were inoculated in 1 L of TB (Merck). The bacteria grew in an incubator at 37 °C, with shaking at 180 rpm. Once the OD reached 2–3, IPTG (300 μM) was added to the media and left overnight at 18 °C. The cultures were harvested by centrifugation. For purification of WT and G177D DNA binding domain (used in 6F58 and 6F59) cell pellets were resuspended in 250 ml of Lysis Buffer (50 mM HEPES pH 7.5, 500 mM NaCl, 10 mM imidazole, 5% glycerol and 1 mM TCEP). The cells, on ice, were sonicated for 20 min with 5 s pulse ON and 10 s pulse OFF with 35% amplitude and centrifuged for 25 min at 66,700 × *g*. The supernatant was incubated for an hour at 4 °C, with nickel beads pre-washed with lysis buffer. After 1 h of batch-binding, the tubes containing the lysate were centrifuged at 700 × *g* at 4 °C for 5 min and the supernatant was discarded. This step was repeated twice with, respectively, 100 ml and 50 ml of lysis buffer. Beads were loaded on a gravity column with 20 ml of wash buffer (50 mM HEPES pH 7.5, 500 mM NaCl, 30 mM imidazole, 5% glycerol, and 1 mM TCEP) and, followed by two elution of 10 ml each with elution buffer (50 mM HEPES pH 7.5, 500 mM NaCl, 300 mM imidazole, 5% glycerol, and 1 mM TCEP). After an SDS-PAGE gel, the elution containing the protein was concentrated with an Amicon 10 kDa concentrator and loaded on a Hi Load 16/600 Superdex 75 pg column at 1 ml/min, collecting 2 ml fractions. The fractions containing the protein were pooled together and concentrated with an Amicon 10 kDa concentrator until 10 mg/ml was reached. Protein aliquots were flash-frozen in liquid nitrogen and stored at −80 °C.

For purification of WT and G177D for DNA-free crystal forms, initial cell lysis and IMAC purification were as above. Following IMAC fractions containing TBXT were pooled, and SUMO protease was added to a final mass ratio 1:150. Cleavage was performed overnight during dialysis into dialysis buffer (50 mM HEPES pH 7.5, 500 mM NaCl, 5% glycerol, 1 mM TCEP) using 3500 MWCO snakeskin dialysis tubing. After dialysis, the protein was concentrated with an Amicon 10 kDa concentrator and loaded on a Hi Load 16/600 superdex 75 pg column. The flow rate of the gel filtration was 1 ml/min and the volume of the fractions collected was 2 ml. The fractions containing the protein were pulled together and concentrated with an Amicon 10 kDa concentrator until the concentration of 12 mg/ml was reached. Protein aliquots were stored at −80 °C after being flash-frozen in liquid nitrogen.

For purification of WT and G177D full-length brachyury (used for SPR analysis), initial cell lysis and IMAC purification were as above. Following IMAC fractions containing TBXT were pooled, and TEV protease was added to a final mass ratio 1:40. Cleavage was performed overnight during dialysis into dialysis buffer (50 mM HEPES pH 7.5, 500 mM NaCl, 5% glycerol, 1 mM TCEP) using 3500 MWCO snakeskin dialysis tubing. After dialysis, the protein was concentrated with an Amicon 30 kDa concentrator and loaded on a Hi Load 16/600 Superdex 200 pg column. All proteins were confirmed by ESI-TOF intact mass spectrometry.

## Crystallization and structure determination

For crystallization of the WT brachyury DNA complex (6F58) a self-complementary DNA oligonucleotide 5′- AATTTCACACCTAGGTGTGAAATT was dissolved to 1 mM, heated to 95 °C on a heat block and allowed to cool slowly over 2 h. The protein and DNA were mixed in a 1:1.1 molar ratio (assuming a duplex DNA molecule) and sitting drop vapor diffusion crystallization trials were set up with a Mosquito (SPT Labtech) crystallization robot at a final concentration of 6.6 mg/ml. TBXT crystallized at 4 °C in conditions containing 40% PEG300, 0.1 M citrate pH 4.2. Crystals were loop-mounted and cryo-cooled by plunging directly into liquid nitrogen. Data were collected to 2.2 Å resolution at Diamond light source beamline I04-1 and the structure was solved by molecular replacement using the program PHASER[44] and the structure of *Xenopus laevis* brachyury[17] (1XBR) as a search model. Refinement was performed using PHENIX REFINE[45] to a final $R_{factor} = 24.2\%$, $R_{free} = 28.8\%$.

For crystallization of the G177D brachyury DNA complex (6F59), a self-complementary DNA oligonucleotide 5′- GAATTTCACACCTAGGTGTGAAATTC was dissolved to 1 mM, heated to 95 °C on a heat block and allowed to cool slowly over 2 h. The protein and DNA were mixed in a 1:1.1 molar ratio (assuming a duplex DNA molecule) and sitting drop vapor diffusion crystallization trials were set up with a Mosquito (SPT Labtech) crystallization robot at a final concentration of 8 mg/ml. TBXT crystallized at 4 °C in conditions containing 56% MPD, 0.1 M SPG pH 6.0. Crystals were loop-mounted and cryo-cooled by plunging directly into liquid nitrogen. Data were collected to 2.1 Å resolution at Diamond light source beamline I04-1 and processed using DIALS[46]. The structure was solved by molecular replacement using the program PHASER[44] and 1XBR[17] as a search model. Refinement was performed using PHENIX REFINE[45] to a final $R_{factor} = 22.1\%$, $R_{free} = 25.2\%$. Data collection and refinement parameters for all DNA complex datasets are shown in Table 1.

## X-ray fragment screening

For crystallization of WT brachyury for fragment screens the protein was adjusted to 7.5 mg/ml and sitting drop vapor diffusion crystallization trials were set up with a Mosquito (SPT Labtech) crystallization. TBXT crystallized at 4 °C in conditions containing 32% PEG400, 0.1 M acetate pH 4.5, 0.1 M cadmium chloride. Crystals were loop-mounted and cryo-cooled by plunging directly into liquid nitrogen.

For crystallization of G177D brachyury for fragment screens the protein was adjusted to 16 mg/ml and sitting drop vapor diffusion crystallization trials were set up with a Mosquito (SPT Labtech) crystallization robot. TBXT crystallized at 4 °C in conditions containing 30% PEG1000, 0.1 M SPG pH 7.0. Initial seed crystals were obtained from these conditions and 4–5 crystals were crushed with a glass probe and transferred to a 50 μl solution of well solution containing a PFTE seed bead and vortexed for 3 × 20 s. The concentrated seed solution was diluted in well solution 1:1000 and used to seed crystals set up at 10 mg/ml in 300 nL drops with 20 nL of seeds (added last).

For the WT crystals, a total of 608 fragments from the DSI poised library[47] (500 mM stock concentration dissolved in DMSO, see Supplementary Data 1) were transferred directly to brachyury crystallization drops using an ECHO liquid handler (10 % or 50 mM nominal final concentration) and soaked for 1–3 h before being loop mounted and flash cooled in liquid nitrogen. A total of 603 crystals were mounted leading to 575 datasets the vast majority of which diffracted to 2.5 Å or higher (96%). For the G177D fragment screen, 637 fragments from the DSI poised library were soaked as above. A total of 590 crystals were mounted leading to 486 datasets with the majority diffracting to 2.5 Å or higher (76%). For both crystal forms data were collected at Diamond light source beamline I04-1 and processed using the automated XChemExplorer pipeline. Structures were solved by difference Fourier synthesis using the XChemExplorer pipeline[48]. Fragment hits were identified using the PanDDA[32] program. Refinement was performed using REFMAC[49]. A view of the electron density maps of each fragment hit is shown in Supplementary Table 2 and the PanDDA event maps in Supplementary Figs. 7 and 8. A summary of data collection and refinement statistics for all fragment-bound and ligand-bound datasets is shown in Supplementary Data 2.

## Electrophoretic mobility shift assays (EMSA)

To evaluate binding to a palindromic DNA sequence the following oligonucleotide pair was used: TA50-F CATGCATGCAGGGAATTTCAC

ACCTAGGTGTGAAATTCCCATTCGTGCGA, TA50-R TCGCACGAATGG GAATTTCACACCTAGGTGTGAAATTCCCTGCATGCATG. To reduce the formation of hairpin structures, the oligos were annealed at a high concentration (>200 μM each) in 10 mM tris- HCl, pH 7.5, 50 mM NaCl by heating to 95 °C in a dry block and leaving to cool to room temperature. The dsDNA was subsequently labeled with T4 polynucleotide kinase (NEB) and γ-32P-ATP. The labeled DNA was separated from the remaining ATP/ADP using a BioRad MicroBiospin P-6 column equilibrated in annealing buffer. EMSA buffer was: 25 mM HEPES, pH 7.4, 10% glycerol, 75 mM NaCl, 0.1% tween 20, and 1 mM TCEP. Protein was diluted serially in this buffer and mixed with 1–5 nM of DNA diluted in the same buffer. After 10-min incubation on ice, the samples (5 μl) were loaded on a pre-run 8% polyacrylamide gel (40:1 acrylamide/bis) in chilled TAE buffer (40 mM TRIS base, 20 mM acetic acid, 1 mM EDTA). The gel tanks were placed in an ice bucket and run for 75 min at 150 V. The dried gels were exposed overnight using a BioRad phosphorimager screen. Results were quantified using BioRad ImageLab software are plotted using Graphpad Prism as means ± standard deviation from at least three independent experiments. Dissociation constants were calculated by fitting the data to a 4-parameter logistic regression equation.

## Differential scanning fluorimetry (DSF)
Full-length WT and G177D brachyury protein was diluted to a final concentration of 2 uM in assay buffer (10 mM HEPES, 150 mM NaCl, 0.5 mM TCEP) containing Sypro Orange (1 in 1000 dilution; Thermo Fisher). Melting curves were obtained from 25 μl samples on an Mx3005p qPCR machine (Agilent), ramping up from 25 to 95 °C, at 1 °C min$^{-1}$. Data were fitted using GraphPad Prism to the Boltzmann equation with Tm values calculated by determining the maximum value of the first derivative of fluorescence transition.

## RNA-seq analysis
RNA-sequencing data was downloaded from GSE121846 and processed using the Genialis platform (https://www.genialis.com). In brief, reads were first pre-processed by BBDuk to remove adapters. Pre-processed reads were aligned to HG19 by HISAT2. For each of the cell lines, aligned reads were examined at the position of amino acid 177 of the mRNA coding region. GGU reads were classified as WT, and GAU reads were classified as containing G177D SNP. For analysis of the CH22 cell line total RNA was sequenced using the DNBSEQ Eukaryotic Strand-specific Transcriptome Resequencing protocol (BGI). FASTQ files were trimmed using Trim Galore (version 0.6.10, Cutadapt version: 4.9, Trimming mode: paired-end, default settings). Trimmed files were aligned using HISAT2 (default settings), and the reference genome used was ENSEMBL GRCh38 version 104. The aligned bam files were sorted and indexed using samtools. Bam files were entered in IGV and the nucleotide composition at chromosome 6 position 166165782 was read out.

## Genotypic chordoma cell lines
In brief, genomic DNA was collected from chordoma cell lines JHC7, UM-Chor1, U-CH1, UM-Chor5, U-CH2, MUG-Chor1, MUG-CC1, U-CH17M, CH22, SW480, and SW620 using the Zymo Quick-DNA™ Miniprep Kit (catalog number D3024). From there, genomic DNA was amplified using primers that flank exon 4 of the TBXT gene (Forward primer 1: CAGAGACACTTTCTTGGGATCCA-GAGGACTT, Reverse primer: TTAGCGCGTCTCCCCGCTCCTCCA). PCR samples were purified using the Zymo DNA Clean & Concentrator-5 kit (catalog number D4004). Samples were sent for Sanger sequencing at Source BioScience. Returned sequencing was examined at the position of amino acid 177 of the DNA. GGT reads were classified at WT, and GAT reads were classified as containing the G177D SNP.

## Surface plasmon resonance (SPR)
DNA binding analysis by SPR was performed on a Biacore s200 machine using a Series S SA sensor surface. Full-length biotinylated G177D or WT brachyury was immobilized to ~1500 RU (immobilization at ~20 nM for 100 s) in running buffer 10 mM Hepes pH 7.5, 150 mM NaCl, 1 mM DTT, 1% DMSO. Palindromic DNA containing T-box binding sites (TA50-F + TA50-R) was titrated as an 8 × 2-fold dilution series from a high concentration of 250 nM in single cycle mode (90 s association with final dissociation of 600 s) with the highest concentration last at a flow rate of 30 μl/min. The data were fit with a bivalent kinetic model (in agreement with each duplex containing two T-box sites) and approximated by dose-response analysis using the Biacore S200 evaluation software.

Small molecule binding by SPR was measured using a Biacore 8K+ system. Full-length human brachyury (G177D) containing a C-terminal AVI tag and biotinylation site was immobilized at ~8000RU to a Biacore SA sensor surface (10 μg/ml for 360 s in buffer 10 mM HEPES, pH 7.5, 150 mM NaCl, 1 mM TCEP, 0.05% P20). Serial dilutions of compounds (6 × 2-fold dilutions starting from 200 μM or 100 μM) were injected over the surface in multi-cycle mode with 60-s association time, 120-s dissociation at a flow rate of 30 ul/min in a buffer containing 10 mM HEPES pH 7.5, 150 mM NaCl, 1 mM TCEP, 0.05 % Tween 20, 1% DMSO. Solvent correction and reference compound injections were performed every 50 and 100 cycles, respectively. After solvent correction compound binding dissociation constants were evaluated using a dose-response (equilibrium) analysis using the Biacore insight evaluation software.

## Reporting summary
Further information on research design is available in the Nature Portfolio Reporting Summary linked to this article.

# Data availability
The crystallographic coordinates and structure factor data generated in this study have been deposited in the Protein Data Bank with the following accession codes: 6F58, 6F59, 8CDN, 5QS6, 5QS7, 5QS8, 5QS9, 5QSA, 5QSB, 5QSC, 5QSD, 5QSE, 5QSF, 5QSG, 5QSH, 5QSI, 5QSJ, 5QSK, 5QSL, 5QRF, 5QRG, 5QRH, 5QRI, 5QRJ, 5QRK, 5QRL, 5QRM, 5QRN, 5QRO, 5QRP, 5QRQ, 5QRR, 5QRS, 5QT0, 5QRT, 5QRU, 5QRV, 5QRW, 5QRX, 5QRY, 5QRZ, 5QS0, 5QS1, 5QS2, 5QS3, 5QS4, 5QS5, 7ZL2, 8A10, 8A7N, 7ZKF, 7ZK2. Ground state datasets used for PanDDA analysis generated in this study are deposited in the Protein Data Bank under accession codes 7HI8 and 7HI9. The SPR sensogram and data fits generated in this study have been made publicly available on Zenodo (https://doi.org/10.5281/zenodo.6394811). Source data are provided with this paper as Source Data file. Source data are provided with this paper.

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

## Acknowledgements

This work was funded in part by grants from the Chordoma Foundation and the Mark Foundation for Cancer Research, and generous gifts to the Chordoma Research Fund by supporters of chordoma patients and chordoma research. The Structural Genomics Consortium (SGC) is a registered charity (number 1097737) that receives funds from Bayer AG, Boehringer Ingelheim, the Canada Foundation for Innovation, Eshelman Institute for Innovation, Genentech, Genome Canada through Ontario Genomics Institute, EU/EFPIA/OICR/McGill/KTH/Diamond, Innovative Medicines Initiative 2 Joint Undertaking, Janssen, Merck KGaA (also known as EMD in Canada and USA), Pfizer, the São Paulo Research Foundation-FAPESP, and Takeda. The crystallographic fragment screen was supported by the XChem facility at Diamond Light Source (proposal ID LB18145). Crystallographic data were collected at Diamond Light Source beamlines I04-1, I04, and I03 (proposals mx19301 and mx28172). We thank the CMD Biotechnology group for help with plasmid cloning, test expression, and mass spectrometry. We thank HD biosciences for assistance with SPR experiments and Piramal for chemistry assistance. P.W. acknowledges recent and current support for his research from Cancer Research UK (CRUK Programme Grants C309/A31322 and C309/A11566; Strategic Award C35696/A23187; Infrastructure Award C309/A27413; and funding for the CRUK Children's Brain Tumour Centre of Excellence C9685/A26398/RG93685), Wellcome (Biomedical Resource and Technology Development Grant 212969/Z/18/Z to support the Chemical Probes Portal), Chordoma Foundation, Mark Foundation, Bone Tumour Research Trust, CRIS Cancer, and The Institute of Cancer Research. P.W. is a CRUK Life Fellow. H.S. was funded by an AACR–CRUK Transatlantic Fellowship. J.A.N. acknowledges support for his research from the following grants NCI P01 CA092584 and CRUK A24759.

## Author contributions

D.H.D., O.G., P.W., P.A.C., C.I.W., and J.A.N.: conceptualization. J.A.N., A.E.G., H.A., N.I., H.E.S., M.A.H., L.T., and Z.W.D.-G.: methodology. J.A.N., A.E.G., H.A., N.I., R.t.P., H.E.S., M.A.H., H.J.O., L.T., and Z.W.D.-G.: investigation and data analysis. J.A.N. and M.A.H.: writing—original draft. D.H.D., O.G., P.W., P.A.C., C.I.W., J.A.N., A.E.G., H.A., N.I., R.t.P., H.E.S., M.A.H., L.T., and Z.W.D.-G.: writing—review and editing final draft.

## Competing interests

P.W. is or has been a consultant/scientific advisory board member for Alterome Therapeutics, Astex Pharmaceuticals, Black Diamond Therapeutics, Charm Therapeutics, CV6 Therapeutics, Cyclacel, EpiCombi AI, Merck KGaA, Nuevolution, Nextech Invest, and Vividion Therapeutics; received grant funding from Nuvectis and Vivan Therapeutics; is a Director of Derwentwater Associates, Storm Therapeutics, and the nonprofit Chemical Probes Portal; holds stock/options in Alterome Therapeutics, Black Diamond Therapeutics, Charm Therapeutics, Chroma Therapeutics, Nextech Invest, and Storm Therapeutics; and is a former employee of AstraZeneca. D.H.D. is on the scientific advisory board of the Chordoma Foundation. He previously served on the Board of Directors for the Chordoma Foundation. The remaining authors declare no competing interests.
