## [Transparent Peer Review file · Nature Communications]

Structural insights into human brachyury DNA recognition and discovery of progressible binders for cancer therapy

Corresponding Author: Dr Joseph Newman

Version 0:

Reviewer comments:

Reviewer #1

(Remarks to the Author)

Newman et al. use the crystal structure of the human transcription factor Brachyury as a starting point to identify small molecular binders using a fragment screening approach. Human Brachyury is a central transcription factor required for notochord and posterior mesoderm development. Aberrant expression of Brachyury is associated with the rare cancer Chordoma making human Brachyury a potential drug target although transcription factors have been considered generally as undruggable in the past.

In the manuscript, the authors solved the crystal structure of wildtype and G177D variant human Brachyury bound to a palindromic and a single DNA binding site as well as in its DNA unbound form. Subsequently, the authors analyzed binding of wildtype and G177D variant to palindromic and single-site DNA, as well as the prevalence of the G177D variant in human Chordoma cell lines. Finally, the authors screened for potential binding pockets and subsequently used a fragment screen of ~600 compounds soaked into crystals containing a further truncated Brachyury construct in its DNA-unbound form. Several hotspots for fragment binding were identified and the bound compounds were further modified using medicinal chemistry ultimately leading to compounds that bind with low μM Kds to human Brachyury.

The workflow and the results presented in this manuscript are interesting and original. Despite the fact that transcription factors are generally considered as undruggable, the authors make a very serious effort to identify high affinity ligands using the combination of fragment screening and medicinal chemistry. The novelty of the manuscript lays mainly in the workflow applied to an "undruggable" transcription factor and the results obtained by fragment screening and the further refinement of these initial hits. In contrast, the structural analysis of the human wildtype Brachyury and the G177D variant as well as the characterization of their DNA-binding properties mainly confirm earlier results, although these analyses are required for the subsequent fragment screening. As also pointed out by the authors, the compounds that have been identified to bind to Brachyury are only starting points because they do not interfere directly with DNA-binding. Instead, the authors suggest that they could be used as PROTAC warheads for protein degradation.

Overall, I consider this a very interesting study. The technical quality of the manuscript is high and the authors present a large amount of data. I therefore recommend publication once the questions below have been addressed.

Specific points:

1.) The EMSAs presented in Fig. 3B are confusing. Is it possible that the "Double" bands the authors observe at the top of the gel are aggregated complex in the loading wells? The authors should therefore show the entire gel. Alternatively, could the presence of tween20 and glycerol lead to the formation of micelles and/or protein aggregation? For me it is unlikely that the authors observe monomer (Single) and dimer (Double) as distinct bands given that there is cooperatively of binding. Is there a way how the authors could further characterize the 'Double' band.

2.) The reasons for the biological relevance of the G177D variant remain unclear. Could it be that this mutation affects interactions with other factors (like other transcription factors). Are there any data that would support this hypothesis? The authors should discuss this point.

3.) The identified fragments only target the protein surface, but not the protein-DNA interface. Is this just a coincidence? In general, how good are the perspectives directly targeting the protein-DNA interface in transcription factors? The authors might discuss this point.

4.) In Table 1, the 3rd column describes a data set "TBXT + ssDNA". What is this data set? I could not find it in the

manuscript.

5.) The Brachyury structure in the absence of DNA has not been described (except in the context of the bound fragments). Is that the structure listed in the 3rd column of Table 1?

Reviewer #2

(Remarks to the Author)

SUMMARY

This communication from Newman and coworkers report the structures and comparative properties of Brachyury, a T-box transcription factor, bound to DNA containing one or two palindromic binding elements, as well as a G177D mutant found in many cases of chordoma, a bone cancer. The structures, together with binding assays, are used to rationalize potential biophysical bases for the pathogenic basis of G177D. The structures are used to execute crystallographic fragment screens for both the WT and G177D proteins. Several binding pockets are identified, but no obvious G177D-specific targets are discerned. Several of the fragments were used to template derivatives that achieve K_d values down to $1e-6$ M. Comprehensive synthesis information for the compounds is provided in SI.

GENERAL COMMENTS

The data is interesting and the extensive suite of structures are valuable. New ligands for "ligandless" transcription factors remain novel and are welcome leads for further development as the authors noted. Having said this, there are major concerns with the manuscript as presented:

The co-crystal structures in Table 1 vary widely in the electron density (2FoFc) coverage for the ligand. Some are certainly very acceptable (e.g., 5QS9), while others are questionable (e.g., 5QS7, 5QSD, 5QSI, 5QRI, 5QRM). This is not a subjective assessment; the Ligand Structure Quality Assessment in the RCSB scores a fair portion of these structures very poorly in terms of fit to experimental data. The discussion around 5QSD in Figure 5b, is problematic for this reason.

Related to the above, there is a lack of mention of structures for most of the structures in Table 2. Are they available? It took unreasonable effort, for instance, to identify the structure for compound 11 (8A7N), which is cursorily mentioned, once, in the list of PDB codes in the Data Availability section. 5QSD, which has poor 2FoFc coverage around the ligand, is used as the reference structure to rationale compound 11, but is unnecessary given 8A7N.

The above items reflect a general issue of pervasive sloppiness and disorganization in the presentation. For example, quantitative parameters such as K_d are missing \pm (whether fitting errors or SD or SE from technical or independent replicates), not a nit-picky issue considering some mediocre fits to the data. Several areas of analysis are also under-developed and require attention, and in some cases, additional data. Specific instances (by section) are raised in the specific comments below.

SPECIFIC COMMENTS

Structural basis of T-box binding element recognition

The discussion is missing attention to the effects of crystal contacts, which are different between the two structures due to the differing space group. It is actually interesting why a point mutant bound to the same DNA could not be crystallized under identical conditions. This should be addressed. The extent to which observed differences may be influenced by differential crystal contacts is therefore important. At least in the WT structure, Gly177 makes direct backbone-backbone contacts with a crystal neighbor. This is not the case for Asp177 in the mutant structure.

Similarly, the DNA:DNA contacts in the palindromic-DNA structures are very different from the corresponding contacts in the single-site structure. Crystal contacts are well known to influence DNA curvature and may well explain any apparent difference in the curvature of the DNA.

The WT (6F58) and G177D (6F59) structures have very different numbers of solvent molecules assigned, much more so than their similar top-end resolutions would suggest. In particular, there is a void near the dyad of the complex where the G177D structure shows no ordered hydration. Either this is the outcome of the wider range of resolution in the mutant structure, or something intrinsic to the structure should be addressed. The relevance of this is highlighted by the author's use of water networks to interpret protein-DNA contacts such as in Figure 1.

Comparison of WT and G177D structures

"Analysis of the interfaces between subunits using the program PISA22 reveals some very minor differences in the interface areas (210 \AA^2 for the G177D versus 219 \AA^2 for the WT) and in the calculated energetic contributions towards toward complex formation (-4.8 Kcal/mol versus -4.5 Kcal/Mol)." The differences are less than 5%, given the atomic resolution, and may be a red herring. To make a statistically informed assessment, the authors should calculate this parameter for different (independent) crystals of each structure.

The WT vs. G177D comparison may benefit from a B'-factor analysis to determine the potential effect of the mutation on flexibility of the relevant loop.

The G177D variant does not substantially change DNA binding affinity or cooperativity
Figure 2A: see comment on crystal contacts above.

The gel in Figure 2B is revealing in that the putative “dimer” band exhibits much lower mobility than the single band. This is not expected unless there are significant conformational differences between the 1:1 and 2:1 complex, or the low-mobility band is actually a higher oligomeric state.

“However contrary to previous reports we do not see any significant differences in either the binding affinity or the degree of cooperativity when comparing the WT and variant proteins.” The notions of “significant” and “substantial” are not well quantified here. Figure 2c clearly suggests the possibility that WT binding is more positively cooperative to the palindromic DNA than for G177D. To be quantitative, the authors should at least fit the data to the Hill equation (if not a mechanistic cooperative model), taking into account ligand depletion if total concentrations are used as independent variables. Then inferential statistics could be done on the relevant parameters to actually state whether WT and G177D differ in cooperativity, given the data. Related to this, the SPR fits in Figure 2D is mediocre, especially for G177D. The effect on the derived parameters should be made at least explicit in the error bars (which are missing).

The Chordoma associated variant is expressed equally to WT
The differential scanning fluorimetry (DSF) method is missing in the methods. What protein concentration is used in Figure 3A? A concentration dependence may be revealing about the point mutation.

“Furthermore, there was no significant difference in the downstream brachyury target genes as identified by ChIP-sequencing” without data or citation of published data. If the latter, provide the GSE ascension. Again, what does “no significant difference” mean in the context of proper statistics for genomic data? Which genes? How many?

Figure 3D: What is UCH-1? This is not mentioned anywhere in the text.

Most importantly, the titular claim that “the Chordoma associated variant is expressed equally to WT” is NOT sufficiently supported by the data provided, which only considers transcriptional output. There may well be differences in transcriptional efficiency of the mRNA and/or metabolic stability (vs thermodynamic stability) of the final protein. Something along the lines of a western blot is clearly in order here.

Crystallographic fragment screening of brachyury

Any misgivings about a potential bias from Cd atoms can be checked by re-screening crystallization conditions of one or more fragment-bound structures.

Many of the sensorgrams in Table S2 are not good quality. If not a more careful redo, the errors in the Kd estimates need to be made explicit.

Structure-guided optimization of fragments to potent brachyury binders

Table 2. *Please* annotate with available co-crystal structures for the compounds.

The text repeatedly mentions Figure S6 but this reviewer cannot make sense of it. The legend to Figure S6 states “showing regions within 4 Å of an atom in a crystal contact” but the text keeps talking about hydrophobicity.

SI

Figure S2 should directly compare with the parameters for the WT-bound structure.

Figure S4 should show mobility shift of for single site next to shift for double sites to provide independent evidence of oligomeric assignments

MISCELLANEOUS

Common nouns should be in lowercase. Ex: “Glycine to Aspartate”, “Epithelial to Mesenchymal” and other instances elsewhere.

8A10 in the Data Availability list is not a relevant structure. If it is a typo, there is a missing code in this list.

Reviewer #3

(Remarks to the Author)

The manuscript by Newman et al presents original work that determines the crystal structure of the human transcription factor brachyury for the first time, comparing the WT and the G177D variant in terms of structure and binding to DNA. The authors also provide some evidence on the biological relevance of the G177D variant. The authors report for the first time the druggability of brachyury and identify potential potent brachyury binders. The manuscript is clear, the results well laid out and I am confident that the work represents an important and significantly contribute to the field.

I would like to offer some specific comments focused on the pathology of brachyury/chordoma and the molecular/cellular experiments presented in the manuscript.

Introduction

Line 64: clarify if duplications are in familial or sporadic chordoma.

Line 72 and 78: I would suggest defining brachyury as the 'oncogenic driver' instead of 'genetic driver' as the majority of sporadic cases of chordoma don't show genetic alterations in the brachyury gene but its expression is likely regulated at the epigenetic level.

Line 28: clarify the sentence 'the malignant phenotype that is expressed almost uniquely in the tumour cells'.

Results

Line 216: The authors claim in this section's title that the chordoma associated G177D variant is expressed equally to the WT, however they mainly show genotyping data and provide limited expression analysis which only partially advances knowledge. Here are some suggestions on how to improve this section and results in Figure 3.

- Line 233: the chordoma associated variant should be described as a SNP (single nucleotide polymorphism)
- Line 227: 'an explanation for observed difference in activity': which activity?
- Figure 3B: this is probably best suited for supplementary data
- Figure 3C: what is the meaning of – and + in the Genotype column? Should this be AA/AG/GG instead? I suggest using the SNP genotyping nomenclature instead as it will help clarify and interpret the data.
- Figure 3D provides very limited evidence about the equal expression of WT/G177D variant transcripts. Could this be improved by:
 - i) Adding profiling of the CH22 or other heterozygous cell lines
 - ii) Perform similar analysis on publicly available transcriptomic data of chordoma patient samples
 - iii) Ideally perform a more sensitive and quantitative analysis of variant expression such as using digital droplet PCR.If the authors want to demonstrate that the two variants are present at similar levels to support the idea that a brachyury-directed chordoma therapy should target both versions of the protein, they should ideally seek to demonstrate the equal expression at the protein level (for example with mass spec), as it is not accurate to infer equal expression by evaluating transcript expression.

Line 225: please define DFS and TM

Methods

I could not find the description of methods used to generate data in Figure 3B-D.

Version 1:

Reviewer comments:

Reviewer #1

(Remarks to the Author)

Overall, the authors have well addressed the different comments of the referees. This is a nice manuscript showing the potential (but also difficulties) using fragment screening to identify small molecules that target transcription factors. I support that the manuscript is now published in Nature Communications.

Reviewer #2

(Remarks to the Author)

The authors have made significant revisions to the text and are commended for their general responsiveness to the comments made in the original review. A residual item of concern relates to Figure 2b, which remains incompletely addressed from the original review and requires further examination by the authors. In Figure 2b, the band marked "double" shows a mobility that is much greater than expected from a homodimer of brachyury subunits. Since electrophoretic mobility scales logarithmically with size, one expects a 2:1 complex to run at lower mobility than 2x the relative mobility of the 1:1 complex ("single"), not the opposite as claimed. In other words, the spacing between the 2:1 and 1:1 should be less than the spacing between 1:1 and unbound. This expected scaling is, in fact, exactly what the authors show for the DNA-binding domains (DBDs) in Figure S5b

While absolute mobilities depend on the details of the gel (matrix composition, buffer, temperature) as the authors noted in the response, relative mobilities should not, unless the hydrodynamic properties of the analytes are themselves also changing due to aggregation, conformational change, etc. Moreover, Figure 2b shows a second unexplained shift in the unbound band (beginning at 11 nM for WT and 22 nM G177D) _after_ the appearance of the "single" bound band and exhibits its own titration.

For these reasons, it is unlikely that the band marked "double" is in fact the 2:1 complex, but rather a higher-order aggregate or even nonspecific binding (since it hardly enters the gel). Given this, the "single" band may also not be the 1:1 complex. The anomalous shift in the unbound band in turn introduces uncertainty in the quantification in Figure 2c, as it is no longer

clear what the disappearance of the unbound band is tracking.

In summary, a better controlled experiment is warranted if gel data were to be used for quantification for full-length brachyury. From visual inspection of Figure S5b, it is already clear that the mutant DBD (G177D) binds more strongly and with higher positive cooperativity than WT (c.f., Figure 2c). The cooperativity comparison is contrary to full-length brachyury based on the present band assignment. (Since the DNA concentration is close to the KD, there is potentially a second confounding factor from titrant depletion if total concentration is taken as the free concentration in the fit. This could directly affect the apparent Hill coefficient.) The reversal, if real, may be interesting (and certainly relevant in view of the discussion in the text), but not supported by the electrophoretic behavior in Figure 2b.

Reviewer #3

(Remarks to the Author)

Thanks for answering my questions.

Regarding Figure 3, if no further evidence is provided, at least I suggest specifying in the subtitle in the main text that this is limited to expression in the cell lines 'The Chordoma associated variant is transcribed equally to WT in cell lines'. Also, I would make it less broad and instead of 'We aimed to understand the consequence of the G177D variant within its endogenous context as other reasons for its association with chordoma could lie in protein stability or expression levels' saying something like: 'We aimed to assess if the WT and G177D variant differ in protein stability or expression levels in cell lines'.

Typo in 'temperature' in Results section.

Add that also CH22 together with the UmChor show equal gene expression in the Results section.

Version 2:

Reviewer comments:

Reviewer #2

(Remarks to the Author)

I appreciate the authors' extensive consideration of their gel mobility shift data. The assignment of slow-moving complexes of full-length brachyury in Figure 2b and attendant conclusions about cooperative binding. If no further evidence is provided, a conservative choice may be to withhold definitive judgement on cooperativity. This does not seem a major loss given the minor difference in overall affinity of the wildtype and G177D brachyury dimer for the palindromic repeat element.

REVIEWER COMMENTS

Reviewer #1 (Remarks to the Author):

Newman et al. use the crystal structure of the human transcription factor Brachyury as a starting point to identify small molecular binders using a fragment screening approach. Human Brachyury is a central transcription factor required for notochord and posterior mesoderm development. Aberrant expression of Brachyury is associated with the rare cancer Chordoma making human Brachyury a potential drug target although transcription factors have been considered generally as undruggable in the past.

In the manuscript, the authors solved the crystal structure of wildtype and G177D variant human Brachyury bound to a palindromic and a single DNA binding site as well as in its DNA unbound form. Subsequently, the authors analyzed binding of wildtype and G177D variant to palindromic and single-site DNA, as well as the prevalence of the G177D variant in human Chordoma cell lines.

Finally, the authors screened for potential binding pockets and subsequently used a fragment screen of ~600 compounds soaked into crystals containing a further truncated Brachyury construct in its DNA-unbound form. Several hotspots for fragment binding were identified and the bound compounds were further modified using medicinal chemistry ultimately leading to compounds that bind with low μM Kds to human Brachyury.

The workflow and the results presented in this manuscript are interesting and original. Despite the fact that transcription factors are generally considered as undruggable, the authors make a very serious effort to identify high affinity ligands using the combination of fragment screening and medicinal chemistry. The novelty of the manuscript lays mainly in the workflow applied to an “undruggable” transcription factor and the results obtained by fragment screening and the further refinement of these initial hits. In contrast, the structural analysis of the human wildtype Brachyury and the G177D variant as well as the characterization of their DNA-binding properties mainly confirm earlier results, although these analyses are required for the subsequent fragment screening. As also pointed out by the authors, the compounds that have been identified to bind to Brachyury are only starting points because they do not interfere directly with DNA-binding. Instead, the authors suggest that they could be used as PROTAC warheads for protein degradation. Overall, I consider this a very interesting study. The technical quality of the manuscript is high and the authors present a large amount of data. I therefore recommend publication once the questions below have been addressed.

We thank the Reviewer for their careful review and overall positive comments and opinion that ‘The novelty of the manuscript lays mainly in the workflow applied to an “undruggable” transcription factor and the results obtained by fragment screening and the further refinement of these initial hits.’ Also their view that ‘Overall, I consider this a very interesting study. The technical quality of the manuscript is high and the authors present a large amount of data.’

Specific points:

1.) The EMSAs presented in Fig. 3B are confusing. Is it possible that the “Double” bands the authors observe at the top of the gel are aggregated complex in the loading wells? The authors should

therefore show the entire gel. Alternatively, could the presence of tween20 and glycerol lead to the formation of micelles and/or protein aggregation? For me it is unlikely that the authors observe monomer (Single) and dimer (Double) as distinct bands given that there is cooperativity of binding. Is there a way how the authors could further characterize the 'Double' band.

This is a valid point we cannot be entirely certain about the identities of the two bands based on this gel alone, although there are several lines of evidence that suggest that the top band is not aggregated protein DNA complex. The wells are not cropped from figure 3B; they would occupy approximately the space where the numbers sit, but they are not visible due to the fact that the gel is imaged via the p32 probe with no clear distinction of the loading wells since none of the probe would be present in this region. It is true that in this particular gel (8 % polyacrylamide) the top band is very close to the position of the wells and could either be a low mobility band or aggregated protein (we estimate it to be within the top 2 mm of the gel). We note that a similar double banding was found in a previous study of brachyury DNA binding (reference 3 in our manuscript) and these bands were also suggested to be monomer and dimer brachyury bound to DNA. Another key piece of evidence that supports our interpretation is that during the course of our study we also examined the binding of the shorter crystallographic constructs (residues 41-244) to this sequence. With these constructs the same type of binding is seen with two bands although in this case both bands are able to enter the gel (consistent with the smaller size of the complex); this time the upper band migrates approximately 1 cm in to the gel. The buffer components tween and glycerol are added actually to suppress this type of aggregation, looking at our gels with the shorter construct there is a small amount of material at the top of the wells that may be aggregation but these are minimal compared to the other bands and we assume this to be the case for the full length brachyury. To make this clearer in the manuscript we have included representative gels of the brachyury (41-244) construct to the Supplementary Information (Figure S5B), and have included the following amendment to the main text (page 5):

“Two shifted species could be observed on the gel for the palindromic probe, the upper band is of low mobility and migrates only minimally into the gel. We assume these two bands represent singly and doubly bound species; this assumption is consistent with experiments using the brachyury DNA binding domain alone (residues 41-224) which show the same banding pattern but both upper and lower bands are able to migrate into the gel distinct from any aggregated proteins that may be present near the wells (Figure S5B).”

2.) The reasons for the biological relevance of the G177D variant remain unclear. Could it be that this mutation affects interactions with other factors (like other transcription factors). Are there any data that would support this hypothesis? The authors should discuss this point.

This is actually our current preferred hypothesis for the role of the variant in chordoma. The most relevant data regarding this is the 2018 study from Beisaw *et al*, in which they identified a brachyury variant Y88A that disrupts the interaction with downstream modulators (in this case the histone acetyltransferase EP300). Y88 and G177 are close to each other in the brachyury structure, and several other disease-associated variants from other T-box family members cluster to this region (roughly identified in our study as “pocket D”) thus linking this site and variant residues within it to possible downstream interactions. We have spent some considerable effort investigating this line of enquiry without any conclusive results, and believe that a definitive answer to this question would require more extensive experiments out of the scope of this manuscript. Since there are conflicting

data in the literature about the impact of the variant on the DNA binding activity *in vitro*, we decided to focus on this aspect and show unequivocally that the variant is equal to WT in this respect. We have now included a brief reference to effectors in the results section of the manuscript on page 7:

“Given the importance of this pocket in other T-Box family members it is tempting to speculate that it plays a role in brachyury for the interaction with its own downstream effectors, possibly explaining the G177D variant association in chordoma.”

3.) The identified fragments only target the protein surface, but not the protein-DNA interface. Is this just a coincidence? In general, how good are the perspectives directly targeting the protein-DNA interface in transcription factors? The authors might discuss this point.

This is an interesting point; the DNA interface is generally accessible in the crystals (an overview of this is shown in Figure S10) but we only found two fragments (entries 5QRW & 5QRM) that bind directly to the DNA interface. The reasons for this are worthy of discussion, there is a classical view in drug discovery that protein DNA interfaces in transcription factors (interfaces with double stranded DNA that is straight or only moderately bent) are by necessity relatively large, flat and polar. These properties are generally not shared by classically druggable sites or pockets known to support binding of small molecules with high affinity. We note that there is relatively little in the way of precedent for directly targeting ligandless transcription factors in the literature (perhaps not unrelated to the challenging nature of these interfaces). We have expanded on this slightly and added the following text to the end of the first paragraph on page 7:

“Across both screens only two fragments (PDB entries 5QRM and 5QRW) are bound to regions containing the DNA binding interface. This is in line with the general view in drug discovery that such interfaces, being relatively flat and polar do not contain tractable pockets to support the binding of small molecules.”

4.) In Table 1, the 3rd column describes a data set “TBXT + ssDNA”. What is this data set? I could not find it in the manuscript.

This is the structure of brachyury in complex with a single T-box element DNA (PDB entry 8CDN); it is mentioned only briefly in the manuscript as it is structurally very similar to the palindromic brachyury DNA complexes (final paragraph of page 3, and second paragraph on page 4) when comparing the binding mode of brachyury on single site versus palindromic DNA (shown on Figure S3). We have added reference to the PDB code and table entry in the text to make this aspect clearer:

“Comparing the structures of brachyury bound to a palindromic DNA with a 12-base pair single site (PDB 8CDN & Table 1, column 3) reveals that most of the contacts at the interface are maintained (Figure S3)”

5.) The Brachyury structure in the absence of DNA has not been described (except in the context of the bound fragments). Is that the structure listed in the 3rd column of Table 1?

The structure in the 3rd column of table 1 is a complex of brachyury with a single T-box element DNA (as above); the brachyury structures without DNA are not really described except for a brief description of their diffraction properties in the 3rd paragraph of page 6, as the proteins are basically

identical to their DNA bound counterparts. We realize this may be slightly confusing and have expanded on this statement (also the description of these structures is useful in reference to Reviewer 2 Specific point 1 which discusses crystal contacts). The description of the APO crystal forms used for the fragment screening experiment now reads:

“WT and G177D brachyury DNA free crystals were obtained in different conditions with different crystal forms although both generally diffracted to around 1.6 Å. The structures of WT and G177D brachyury in the absence of DNA are almost identical to their counterparts bound to DNA (R.M.S.D. 0.36 and 0.37 respectively), including the significant differences observed around the G177D variant despite different crystal packing interactions, thus validating that these differences are not crystallographic artefacts (Figure S4).”

We have also deposited APO (Ground state) versions of these structures (PDB codes 7HI8 and 7HI9) and added the statistics for these entries to Table I.

Reviewer #2 (Remarks to the Author):

SUMMARY

This communication from Newman and coworkers report the structures and comparative properties of Brachyury, a T-box transcription factor, bound to DNA containing one or two palindromic binding elements, as well as a G177D mutant found in many cases of chordoma, a bone cancer. The structures, together with binding assays, are used to rationalize potential biophysical bases for the pathogenic basis of G177D. The structures are used to execute crystallographic fragment screens for both the WT and G177D proteins. Several binding pockets are identified, but no obvious G177D-specific targets are discerned. Several of the fragments were used to template derivatives that achieve Kd values down to 1e-6 M. Comprehensive synthesis information for the compounds is provided in SI.

GENERAL COMMENTS

The data is interesting and the extensive suite of structures are valuable. New ligands for “ligandless” transcription factors remain novel and are welcome leads for further development as the authors noted. Having said this, there are major concerns with the manuscript as presented:

We thank the Reviewer for their positive comment that the data are interesting and their view that ‘New ligands for “ligandless” transcription factors remain novel and are welcome leads for further development as the authors noted.’

The co-crystal structures in Table 1 vary widely in the electron density (2FoFc) coverage for the ligand. Some are certainly very acceptable (e.g., 5QS9), while others are questionable (e.g., 5QS7, 5QSD, 5QSI, 5QRI, 5QRM). This is not a subjective assessment; the Ligand Structure Quality Assessment in the RCSB scores a fair portion of these structures very poorly in terms of fit to experimental data. The discussion around 5QSD in Figure 5b, is problematic for this reason.

We do not dispute or are in any way trying to hide the observations of the Reviewer. It is established that the PANDDA algorithm is able to find binders in electron density maps that are not possible to

find by conventional difference map analysis (discussed in more detail in reference 32). In our experience of this technique over several targets that diffract to similar resolution, we find roughly one third of fragments have convincing density in 2FoFc maps, another third of fragments have 2FoFc maps with good density at lower contour levels, whilst the final third of fragments do not have good density in 2FoFc maps (except at very low contour levels) and would not be found if using a conventional difference map analysis. It is assumed that the former are high occupancy and the latter lower occupancy fragments. This pattern is indeed the same for the brachyury fragments in this manuscript, some of which (as pointed out by the reviewer) would not be identifiable in a conventional 2FoFc map. For these fragments, the main evidence for their binding pose comes from the PANDDA event maps which are calculated from a statistical analysis of multiple APO structures and feature a background subtraction procedure that highlights the electron density of low occupancy bound states to allow model building. This procedure has become standard practice for high-throughput X-ray fragment screening datasets and has been validated and published independently of this study (For example reference 32). It is regretful that the event maps are not more easily accessible, the PDB only accept raw diffraction data and coordinates for deposition (at least this is the case when the data in this paper were deposited). There are efforts to improve this situation, with it being possible to recreate these maps in theory, but it is difficult to do for non-experts. In light of this we have included two new figures in the Supplemental data (Figures S7 & S8) which actually show the event maps for each fragment in this study. We have also deposited to the PDB ground state models for the PANDDA analysis that contain all the necessary data to recompute these maps. These data should allow the reader to see the evidence for the fragment (event map) alongside the more traditional 2FoFc maps in Supplemental Table S2 which serve as a useful comparative measure allowing differentiation of the stronger high occupancy and weaker low occupancy hits. The 5QSD fragment in question is one of the weaker hits although it does have good event maps and reasonable density in 2Fo-Fc maps at lower contour. We have added the following to the results section to make these points clearer:

“In both fragment screens hits were identified ranging from high occupancy ligands with good quality electron density, to low occupancy ligands with only weak density if analysed using a conventional 2Fo-1Fc electron density map (Table S2). For these lower occupancy ligands, the main evidence for ligand binding comes from the PanDDA event maps which are shown in Figures S7 & S8.”

Related to the above, there is a lack of mention of structures for most of the structures in Table 2. Are they available? It took unreasonable effort, for instance, to identify the structure for compound 11 (8A7N), which is cursorily mentioned, once, in the list of PDB codes in the Data Availability section. 5QSD, which has poor 2FoFc coverage around the ligand, is used as the reference structure to rationale compound 11, but is unnecessary given 8A7N.

We apologize for this, we have made efforts to relate the fragment structures to PDB codes (for example Table S2) although we regret that this includes only the original fragment screening datasets and not structures of follow up compounds (such as 8A7N or compound 11). We were not able to obtain structures for many of the compounds in Table II, we have discussed this in the manuscript (see Results section final paragraph) with the insight that the cyclopropyl moiety which appears to be favourable for binding is no longer compatible with soaking into the existing crystal forms. We have added references to PDB codes 7ZK2 and 8A7N in the legend to Figure 5 and to Table 2, to better aid the reader to identify these structures.

As for the discussion about the 5QSD fragment (related to the previous point), 5QSD was the original (weak) hit that inspired the design of the compound series that compound 11 belongs to. This point surely validates that even weak binders contain valuable information that may be used to progress towards more potent binders as we demonstrate in this instance.

The above items reflect a general issue of pervasive sloppiness and disorganization in the presentation. For example, quantitative parameters such as K_d are missing \pm (whether fitting errors or SD or SE from technical or independent replicates), not a nit-picky issue considering some mediocre fits to the data. Several areas of analysis are also under-developed and require attention, and in some cases, additional data. Specific instances (by section) are raised in the specific comments below.

We apologise for any general sloppiness in the manuscript and have taken steps to correct some of these issues ourselves. The points about error estimates for K_d values and error bars for replicates are well made and we have corrected this in the revised version of the manuscript.

SPECIFIC COMMENTS

Structural basis of T-box binding element recognition

The discussion is missing attention to the effects of crystal contacts, which are different between the two structures due to the differing space group. It is actually interesting why a point mutant bound to the same DNA could not be crystallized under identical conditions. This should be addressed. The extent to which observed differences may be influenced by differential crystal contacts is therefore important. At least in the WT structure, Gly177 makes direct backbone-backbone contacts with a crystal neighbor. This is not the case for Asp177 in the mutant structure.

One important piece of information regarding this is the structure of the DNA-free WT and G177D crystals used for fragment screening. Again the two variants crystallized in different space groups (perhaps not unsurprising given the context of the surrounding structural changes on a relatively small protein) and have different crystal contacts. Importantly the differences observed in the loops surrounding G177 are preserved and thus the differences are observed independently of crystal contacts. We have added the following to page 6:

“WT and G177D brachyury DNA free crystals were obtained in different conditions with different crystal forms although both generally diffracted to around 1.6 Å. The structures of WT and G177D brachyury in the absence of DNA are almost identical to their counterparts bound to DNA (R.M.S.D. 0.36 and 0.37 respectively), including the significant differences observed around the G177D variant despite different crystal packing interactions, thus validating that these differences are not crystallographic artefacts (Figure S4A & S4B).”

Similarly, the DNA:DNA contacts in the palindromic-DNA structures are very different from the corresponding contacts in the single-site structure. Crystal contacts are well known to influence DNA curvature and may well explain any apparent difference in the curvature of the DNA.

We note again that the WT and G177D structures bound to palindromic DNA, PDBid's 6F58 and 6F59 respectively, are also very different from each other in the DNA packing. The 6F58 structure contains DNA end-to-end contacts that form a pseudo-contiguous DNA double helix. This is not the case for

6F59 which does not contain significant DNA to DNA crystal contacts. The single site DNA structure 8CDN is similar to 6F58 in that it contains DNA end-to-end crystal contacts. A similar argument to the above is relevant in this case, as the differences we observe in DNA conformation are present in both the 6F58 and 6F59 structures. If these differences were due to crystal contacts they would not be present for both structures, since by definition crystal contacts effects are specific to a particular crystal form. We have added the following text to help clarify this point, and have also added a Supplemental Figure (Figure S4B) showing the crystal contacts for DNA-free and DNA complex structures and included both 6F58 and 6F59 in the revised version of Figure S3:

“These distortions are present across both the WT and G177D DNA-bound structures which have different crystal contacts and DNA to DNA interactions (Figure S4B), indicating that these are not a result of the DNA environment. “

The WT (6F58) and G177D (6F59) structures have very different numbers of solvent molecules assigned, much more so than their similar top-end resolutions would suggest. In particular, there is a void near the dyad of the complex where the G177D structure shows no ordered hydration. Either this is the outcome of the wider range of resolution in the mutant structure, or something intrinsic to the structure should be addressed. The relevance of this is highlighted by the author’s use of water networks to interpret protein-DNA contacts such as in Figure 1.

This is an astute observation but we are not sure that it indicates a problem in need of fixing. We have searched the literature for information on expected number of waters per residue as a function of resolution and the most relevant literature we could find is a paper by Bernhard Rupp (PMID 2594568) which displays the number of water molecules per residue gathered from PDB statistics in graphical form in Figure 6 (plotted as a function of resolution). We note in this graph that there is large variance at all but the highest resolutions including around the 2.1-2.2 Å range of our current structures. The 6F58 structure by this metric contains 0.87 waters per residue and sits within the upper whisker (defined as 1.5 times the inner quartile range), whilst the 6F59 structure contains 0.29 waters per residue (within the grey box which signifies the first quartile). The median from this analysis appears to be around 0.5 water per residue at 2.1-2.2 Å. From this analysis whilst the two structures are in the upper and lower range, they are not outliers in the distribution.

We can only speculate as to the reason behind this. Our best guess is that it may be related to the differences in the Wilson B factor for the two crystal forms. A similar nominal resolution can be achieved by a small crystal with high internal order (this is approximated by the Wilson B factor which describes mathematically the drop-off in diffraction intensity versus resolution) or a larger crystal with a lower degree of internal order. The Wilson B factor of 6F58 is 34 whilst 6F59 is 59. Consistent with this the 6F58 crystals were slim and plate like, whilst the 6F59 crystals were larger and more 3D/chunky. It can be assumed that the actual number of waters coordinated in the hydration shell of proteins is fairly constant but the ability to see these in electron density maps is a function of how well ordered they are. In the Rupp paper (referred to above) the B-factors of water molecules are discussed and observed to generally approach or exceed the B-factors of their environmental neighbours (but not lower), thus a structure with higher B-factors may reasonably be expected to contain fewer ordered waters per residue as a result of the waters having a B-factor higher than what can reliably be modelled in an electron density map, even at the same diffraction resolution. As the number of waters in our DNA bound structures fits roughly within the observed

statistical distributions for other structures in the PDB and the argument above is largely speculative we have not included these observations within the manuscript.

Comparison of WT and G177D structures

“Analysis of the interfaces between subunits using the program PISA22 reveals some very minor differences in the interface areas (210 Å² for the G177D versus 219 Å² for the WT) and in the calculated energetic contributions towards toward complex formation (-4.8 Kcal/mol versus -4.5 Kcal/Mol).” The differences are less than 5%, given the atomic resolution, and may be a red herring. To make a statistically informed assessment, the authors should calculate this parameter for different (independent) crystals of each structure.

We are generally in agreement with the reviewer here. Our assessments of these differences are noted as “very minor” and we did not make any statements about the significance of these differences. We are not quite sure what the suggestion to calculate with independent crystals implies, presumably using additional datasets of each complex. We do not actually have such data and are somewhat doubtful as to their value as being truly independent. We would expect such structures to be virtually identical, in fact this is the case for our fragment screening structures for which we have collected data from many hundreds of crystals and structures generally align with RMSD values close to the coordinate error ~0.25 Å. We do acknowledge that this variance may be higher in the lower resolution crystals of the DNA bound form. We have added the following statement to the results section on page 4 to make the point that we consider the differences minor more explicit:

“Analysis of the interfaces between subunits using the program PISA22 reveals only very minor differences in the interface areas (210 Å² for the G177D versus 219 Å² for the WT) and in the calculated energetic contributions towards toward complex formation (-4.8 Kcal/mol versus -4.5 Kcal/Mol); we consider these differences to be unlikely to have a role in the biological activity of brachyury.”

The WT vs. G177D comparison may benefit from a B'-factor analysis to determine the potential effect of the mutation on flexibility of the relevant loop.

This is a good point; we have added the following to the discussion on the loop:

“The crystallographic B-factors indicate this loop is fairly mobile in both the WT and G177D brachyury structures with B-factors slightly higher than neighbouring residues for both structures, although the potential stabilizing influence of crystal contacts cannot be ruled out (Figure S4A & S4B).”

The G177D variant does not substantially change DNA binding affinity or cooperativity
Figure 2A: see comment on crystal contacts above.

We have addressed this point above with observations that confirm structural differences across different crystal forms with different crystal contacts.

The gel in Figure 2B is revealing in that the putative “dimer” band exhibits much lower mobility than

the single band. This is not expected unless there are significant conformational differences between the 1:1 and 2:1 complex, or the low-mobility band is actually a higher oligomeric state.

We are not sure that we follow the Reviewer's argument here. Migration on a native polyacrylamide gel is a function of the charge of the species and its size. The DNA alone is a 50 bp duplex of molecular mass of approximately 30 kDa whilst the singly bound species has a theoretical molecular mass of 80 kDa and the doubly bound species 130 kDa. We would expect the 130 kDa species to migrate slower than the 80 kDa even if in the same conformational state. The extent of this difference is highly dependent on the acrylamide percentage and gel running conditions (voltage, current and duration) of the gel. We are not able to say that the observed pattern we see is necessarily any different from what would be expected.

The identities of these gel bands were also questioned by Reviewer 1 and we do see some merit in this argument. We have (as detailed above) included some additional data (Figure S5B) on a truncated brachyury construct (the same as used in crystallization of DNA complexes) which shows a similar banding pattern although more mobile within the gel and the bands are slightly closer together consistent with a smaller mass difference (52 kDa for single and 74 kDa) for the single and double DNA bound versions of this construct.

"However contrary to previous reports we do not see any significant differences in either the binding affinity or the degree of cooperativity when comparing the WT and variant proteins." The notions of "significant" and "substantial" are not well quantified here. Figure 2c clearly suggests the possibility that WT binding is more positively cooperative to the palindromic DNA than for G177D. To be quantitative, the authors should at least fit the data to the Hill equation (if not a mechanistic cooperative model), taking into account ligand depletion if total concentrations are used as independent variables. Then inferential statistics could be done on the relevant parameters to actually state whether WT and G177D differ in cooperativity, given the data. Related to this, the SPR fits in Figure 2D is mediocre, especially for G177D. The effect on the derived parameters should be made at least explicit in the error bars (which are missing).

This is a valid point; our observations mainly relate to a visual assessment of the gels in Figure 2B rather than the Hill slope of the quantification in Figure 2C. To clarify these data were quantified by combining bands corresponding to the upper and lower shifted species, and fit to a 4-parameter logistic equation that includes a Hill slope. The calculated Hill coefficient were 2.285 for the WT and 1.832 for the G177D mutant. For this type of fit, errors in parameters are best described in terms of a 95 % confidence interval (due to the asymmetric nature of the error distribution). For the Hill slopes the 95% confidence intervals are between 1.94 – 2.73 for the WT, and 1.45 to 2.34 for the G177D mutant. These distributions overlap and would not be considered significantly different if using a traditional measure of significance (roughly equivalent to a P-value of <0.05). We apologize that some of the statistical analysis was not included in the manuscript and have now added the Hill slopes to the Results section on page 5 and added a description of the error bars to the figure legend Figure 2c:

"However, contrary to previous reports, we do not see any significant differences in either the binding affinity or the degree of cooperativity when comparing the WT and variant proteins, with the dissociation constants [1.4 nM WT (95%CI 1.32 to 1.53) and 1.2 nM G177D (95%CI 1.03 to 1.35)] and Hill slopes [2.3 WT (95%CI 1.93 to 2.74) and 1.8 G177D (95%CI 1.46 to 2.35)] from the quantification of the data being not significantly different if using a 95% confidence interval (Figure 2C)"

Given that the gel images in Figure 2b appear visually almost identical we do not feel that a more sophisticated analysis of cooperative binding is warranted. Furthermore, we provide an independent orthogonal measurement via SPR that strengthens this conclusion. We do not really agree with the Reviewer's assessment of the fits to the SPR data as mediocre; however we have displayed very clearly these data to allow the reader to judge for themselves. Whilst some of the data quality may be questioned, we believe our conclusions that the variant and WT do not appear different are justified even in the case of the fits. For clarity we have added a table in the Supplemental Data that includes the parameters and errors on the SPR fits (Supplemental Table S1A & S1B).

The Chordoma associated variant is expressed equally to WT

The differential scanning fluorimetry (DSF) method is missing in the methods. What protein concentration is used in Figure 3A? A concentration dependence may be revealing about the point mutation.

This was an error on our part, for which we apologize. We have included these data on the revised manuscript (Materials and methods section). Both the WT and mutant were analysed at the same 2 uM protein concentration.

"Furthermore, there was no significant difference in the downstream brachyury target genes as identified by ChIP-sequencing" without data or citation of published data. If the latter, provide the GSE ascension. Again, what does "no significant difference" mean in the context of proper statistics for genomic data? Which genes? How many?

This relates to publicly available data not included in this manuscript. We apologize for the missing citation and have clarified this to the revised version of the manuscript and included details of the GSO ascension number (GSE109794, Sheppard et al, 2021). We are not trying to reinterpret or analyse that data and are only quoting from the conclusions in those papers.

Figure 3D: What is UCH-1? This is not mentioned anywhere in the text.

Apologies this is a typo for U-CH1 a sacral chordoma cell line provided by the Chordoma Foundation that is homozygous for the G177D variant, we have corrected this typo in the Figure and legend.

Most importantly, the titular claim that "the Chordoma associated variant is expressed equally to WT" is NOT sufficiently supported by the data provided, which only considers transcriptional output. There may well be differences in transcriptional efficiency of the mRNA and/or metabolic stability (vs thermodynamic stability) of the final protein. Something along the lines of a western blot is clearly in order here.

These are valid points, but we are not able to investigate all of these possibilities in the current paper. We have tried to focus on our strengths in biochemical and biophysical characterization and have tested the DNA binding activity *in vitro* and the thermodynamic stability. We acknowledge that other explanations are possible. A western blot could be informative but we are not aware of any antibodies that are able to discriminate WT from G177D variant brachyury. As with the suggestion of Reviewer 3, answering this question fully would likely require a large and detailed proteomics (mass spectroscopy) type approach across many cell lines and tissue samples that we believe to be beyond

the scope and allowed length of the current study. To reflect these points we have changed the section title to “the Chordoma associated variant is transcribed equally to WT” and added the following note on the metabolic stability:

“examining the stability of WT and G177D brachyury by DSF shows only a small TM shift of ~0.7 degrees, with the WT variant appearing to be very slightly more thermostable than the chordoma-associated variant (Figure 3A) and thus likely not an explanation for observed differences in activity, although we cannot rule out differences in transcriptional efficiency or metabolic stability due interactions with cellular degradation machinery.”

Crystallographic fragment screening of brachyury

Any misgivings about a potential bias from Cd atoms can be checked by re-screening crystallization conditions of one or more fragment-bound structures.

We have tried quite extensive screening of possible conditions; the Cd ions are necessary for the crystallization of the WT brachyury (DNA-free form) and we are unable to get high resolution diffraction from crystals for this construct in other conditions. As the fragments were soaked into the crystals we are not able to try this.

Many of the sensorgrams in Table S2 are not good quality. If not a more careful redo, the errors in the Kd estimates need to be made explicit.

We accept that the quality of fits is mixed with some better than others, as is often the case with follow up SPR studies. As indicated elsewhere, we have displayed the data clearly and allow the reader to judge. We apologize for the omission of the errors in Kd estimates which we acknowledge should have been included and have now added these to the revised version.

Structure-guided optimization of fragments to potent brachyury binders

Table 2. *Please* annotate with available co-crystal structures for the compounds.

This has been added.

The text repeatedly mentions Figure S6 but this reviewer cannot make sense of it. The legend to Figure S6 states “showing regions within 4 Å of an atom in a crystal contact” but the text keeps talking about hydrophobicity.

Apologies, there is a typo in the text that refers to a hydrophobic cluster of residues which should refer to what is now Figure S10. We have corrected this in the revised version.

SI

Figure S2 should directly compare with the parameters for the WT-bound structure.

This is a fine suggestion and we have added these parameters for the WT (6f58) structure in the revised document.

Figure S4 should show mobility shift of for single site next to shift for double sites to provide

independent evidence of oligomeric assignments

We have included additional data to this figure, which we have detailed above to support our assessments.

MISCELLANEOUS

Common nouns should be in lowercase. Ex: "Glycine to Aspartate", "Epithelial to Mesenchymal" and other instances elsewhere.

We apologize for these errors and have corrected them in the revised versions.

8A10 in the Data Availability list is not a relevant structure. If it is a typo, there is a missing code in this list.

Again apologies, this was a typo, the PDB code should be 8A10 not 8A1O (zero not o). We thank the reviewer for their attention to detail and have corrected this in the revised manuscript.

Reviewer #3 (Remarks to the Author):

The manuscript by Newman et al presents original work that determines the crystal structure of the human transcription factor brachyury for the first time, comparing the WT and the G177D variant in terms of structure and binding to DNA. The authors also provide some evidence on the biological relevance of the G177D variant. The authors report for the first time the druggability of brachyury and identify potential potent brachyury binders. The manuscript is clear, the results well laid out and I am confident that the work represents an important and significantly contribute to the field. I would like to offer some specific comments focused on the pathology of brachyury/chordoma and the molecular/cellular experiments presented in the manuscript.

We thank the Reviewer for highlighting that 'The manuscript by Newman et al presents original work that determines the crystal structure of the human transcription factor brachyury for the first time, comparing the WT and the G177D variant in terms of structure and binding to DNA.' We also appreciate that the Reviewer points out that 'The authors report for the first time the druggability of brachyury and identify potential potent brachyury binders and that' The manuscript is clear, the results well laid out and I am confident that the work represents an important and significantly contribute to the field.'

Introduction

Line 64: clarify if duplications are in familial or sporadic chordoma.

We have clarified this point, Familial chordoma always contains a germline tandem duplication of brachyury whilst sporadic chordoma duplications are more infrequent (27% of cases). This section now reads:

"The *TBXT* (*brachyury*) gene is always duplicated in rare familial chordoma and in some sporadic chordomas⁸ (27% of cases). It is required for growth in chordoma"

Line 72 and 78: I would suggest defining brachyury as the 'oncogenic driver' instead of 'genetic driver' as the majority of sporadic cases of chordoma don't show genetic alterations in the brachyury gene but its expression is likely regulated at the epigenetic level.

We thank the reviewer for this clarification, we have changed "genetic driver" to "oncogenic driver" as suggested.

Line 28: clarify the sentence 'the malignant phenotype that is expressed almost uniquely in the tumour cells'.

This is to emphasize that brachyury is not expressed in healthy adult tissues (with the exception of the thyroid, testes and pituitary gland). We have rephrased the sentence to make the meaning clearer:

"Overall, these data are consistent with brachyury being the oncogenic driver in chordoma that is minimally expressed in healthy adult tissues, making it a biologically ideal therapeutic target in chordoma."

Results

Line 216: The authors claim in this section's title that the chordoma associated G177D variant is expressed equally to the WT, however they mainly show genotyping data and provide limited expression analysis which only partially advances knowledge. Here are some suggestions on how to improve this section and results in Figure 3.

- Line 233: the chordoma associated variant should be described as a SNP (single nucleotide polymorphism)

We have corrected this, SNV is now changed to SNP, the abbreviation is also defined in the introduction Paragraph 2.

- Line 227: 'an explanation for observed difference in activity': which activity?

We agree this was not clear, the use of "activity" is merely the statistical association of this variant with chordoma (odds ratio of 6.1). We have clarified this sentence:

"likely not an explanation for the association of the G177D SNP with chordoma".

- Figure 3B: this is probably best suited for supplementary data

Arguably this is true although we think that the panel B is useful in reference to the interpretation of the genotype data in panel 3C and would prefer to keep these data together.

- Figure 3C: what is the meaning of – and + in the Genotype column? Should this be AA/AG/GG instead? I suggest using the SNP genotyping nomenclature instead as it will help clarify and interpret the data.

We agree that this needed clarifying, in the column + is used to signify the WT allele and – the G177D mutant. We have now, as the reviewer suggested changed this to AA/AG/GG and added more details to the figure legend.

- Figure 3D provides very limited evidence about the equal expression of WT/G177D variant transcripts. Could this be improved by:

- i) Adding profiling of the CH22 or other heterozygous cell lines
- ii) Perform similar analysis on publicly available transcriptomic data of chordoma patient samples
- iii) Ideally perform a more sensitive and quantitative analysis of variant expression such as using digital droplet PCR.

These are all excellent suggestions. Unfortunately, the PostDoc (H.S.) who Performed the original RNA-Seq analysis has started a new role in industry and no longer has access to the software or tools necessary to analyse this data. We thus enlisted the help of a colleague and now new co-author (R.T.P) who assisted us to analyse the RNA-seq data from the U-CH2 heterozygous cell line. As shown in Figure 3 the CH22 cell line also has equal RNA-seq reads for the WT and G177D allele (49.41% and 50.57 % respectively out of a total of 7900 reads with a standard deviation of 0.73). This is in agreement with our original analysis and adds support to our claims for equal transcriptional output of these variants in chordoma.

If the authors want to demonstrate that the two variants are present at similar levels to support the idea that a brachyury-directed chordoma therapy should target both versions of the protein, they should ideally seek to demonstrate the equal expression at the protein level (for example with mass spec), as it is not accurate to infer equal expression by evaluating transcript expression.

We accept the point (also raised by Reviewer 2), that we only show transcriptional output and not data on protein levels or the relative roles in chordoma. We have changed our wording in the Results Section to be more accurate. As suggested by the Reviewers, proteomics-based approaches would be the best way to truly answer this question. However, especially given the focus of the present paper, we think that our transcriptomics analysis and the genotype of the U-CH17M cell line (heterozygous to the WT allele) are evidence enough to support the general conclusion we draw, which is now expressed as that an effective chordoma therapeutic targeting brachyury “will most likely be needed to target both the WT and G177D variants” to indicate that the conclusion is not definitive. To properly address this question of the relative protein levels of WT vs G177D brachyury in chordoma would require a large-scale proteomics approach using multiple cell lines and human tumour tissue samples. This would be a very big undertaking that we feel is well beyond the scope and allowed length of the present manuscript, which is focussed very heavily on publishing for the first time the crystal structures of DNA-free and DNA-bound human brachyury, both the WT and the G177D variant, and on and on discovery of small molecule brachyury binders. In further response to the Reviewers we have indicated the future need to carry out mass spec-based proteomic analysis of the relative expression WT and G177D variant in multiple cell lines and patient samples. We have been advised by the Chordoma Foundation that such proteomic profiling studies are already underway and the data sets will be published in due course. In addition, in the final section we have

made the point that future PROTACs, potentially derived from our lead compounds and binding to either or both forms, could be used to determine their relative involvement in the development and growth of chordoma models – and hence inform on the best therapeutic strategy.

Line 225: please define DFS and TM

We have added these definitions and also details of this experiment to the Methods sections.

Methods

I could not find the description of methods used to generate data in Figure 3B-D.

Apologies, these were omitted in error and have now been added.

Reviewer #1 (Remarks to the Author):

Overall, the authors have well addressed the different comments of the referees. This is a nice manuscript showing the potential (but also difficulties) using fragment screening to identify small molecules that target transcription factors. I support that the manuscript is now published in Nature Communications.

We thank the reviewer once again for their careful review and supportive comments.

Reviewer #2 (Remarks to the Author):

The authors have made significant revisions to the text and are commended for their general responsiveness to the comments made in the original review. A residual item of concern relates to Figure 2b, which remains incompletely addressed from the original review and requires further examination by the authors.

We thank the reviewer for the commendation of our attempts to respond to their comments, we will attempt to further clarify this concern.

In Figure 2b, the band marked “double” shows a mobility that is much greater than expected from a homodimer of brachyury subunits. Since electrophoretic mobility scales logarithmically with size, one expects a 2:1 complex to run at lower mobility than 2x the relative mobility of the 1:1 complex (“single”), not the opposite as claimed.

We have some confusion about the reviewer’s comments above, in the original review the reviewer states that “that the putative “dimer” band exhibits much lower mobility than the single band. This is not expected...” this is in contradiction to the above that claims the double band mobility is greater than expected. We presume based on the rest of the argument and the comparisons to Figure S5b that the reviewer would expect a 2:1 complex to enter the gel and migrate somewhat closer to the single band than what we show. We have based our response below on this assumption but please accept our apologies if this is not the case.

In other words, the spacing between the 2:1 and 1:1 should be less than the spacing between 1:1 and unbound. This expected scaling is, in fact, exactly what the authors show for the DNA-binding domains (DBDs) in Figure S5b

We note that this is a native 12 % TBE gel of a protein DNA complex and mobilities depend both on the charge and hydrodynamic properties. The charges of the various complexes are not necessarily known and these various complexes almost certainly have different mass to charge ratios (see below), we do not think that making any predictions about the relative mobilities of bands in this case is especially valid.

While absolute mobilities depend on the details of the gel (matrix composition, buffer, temperature) as the authors noted in the response, relative mobilities should not, unless the hydrodynamic properties of the analytes are themselves also changing due to aggregation, conformational change, etc.

Again, we do not agree with this point, both absolute and relative mobilities do vary among different gel compositions (acrylamide %, buffering ions and pH). Even in the simplified case of an SDS page gel (where mass to charge ratio is constant) the relative mobilities of different bands vary depending on the acrylamide/agarose percentage. A simple example of this is protein marker profile on an SDS PAGE gels, with higher agarose/acrylamide concentrations giving greater separation of smaller species whilst lower agarose resolving larger. We show below an example to illustrate this showing the migration of commercial markers approximating the sizes of the species of the bands under scrutiny where the relative mobility (calculated as the mobility of a smaller band as a multiple of the largest band) varies clearly according to the acrylamide percentage of the gel matrix.

Molecular weight kDa	15 % Acrylamide gel		8 % Acrylamide gel	
	Migration %	Relative Mobility	Migration %	Relative Mobility
130	6.5	1	21	1
70	14	2.15	39	1.56
35	38	5.84	75	3.57

Migration values estimated from figure on left.

Relative mobility expresses the migration of the 70 kDa and 35 kDa bands relative to the 130 kDa

Image adapted from Thermo scientific

Our EMSA gels have a further source of variability that can cause species to migrate differentially in that the mass to charge ratio of the complexes is no longer constant. The charge of the complexes may be difficult to calculate accurately but a simplistic approach is to count the net number of positive or negative charges of the species based on its chemical composition. We have tabulated below the mass to charge ratio of the identities of all the bands under discussion.

Assumed Species	Mass (Da)	Net Charge	Mass to Charge m/z (amu)
DNA (50mer)	30770	-100	-307.7
DNA + Brachyury (monomer)	78283	-103	-760.0291262
DNA + Brachyury (dimer)	125796	-106	-1186.754717
DNA + DBD (monomer)	52815	-96	-550.15625
DNA + DBD (dimer)	74860	-92	-813.6956522

This table demonstrates that the theoretical mass to charge ratio of the complexes are different and may help to reframe the expected migration of these species on a gel. Within this framework we do not agree that the migration of the double band is any different than expected as it is significantly larger than any other band and has greater mass to charge ratio. We do accept the argument that the migration of the presumed monomer band in Figure 2b is faster than expected compared to the DNA + DBD (dimer) band in figure 55b as the two species are of similar masses and m/z ratios. We are not able to fully explain this; however, the experiments were performed independently on different gels making a direct comparison potentially misleading. We also note that the overall charge and PI of the DNA binding domain alone construct is positive with a PI of 9.0 and the overall charge of the full length brachyury is negative with PI of 6.6 potentially confounding their migration patterns in the gel.

Moreover, Figure 2b shows a second unexplained shift in the unbound band (beginning at 11 nM for WT and 22 nM G177D) _after_ the appearance of the “single” bound band and exhibits its own titration.

Whilst we cannot be entirely sure we presume that this band represents partial dissociation of the DNA protein complex that has occurred over the timescale of the electrophoresis. Such patterns are not uncommon and are a limitation of the EMSA experiment that these samples are not in chemical equilibrium during the electrophoresis step. The dissociation of the complex over time gives a mobility that is intermediate to the bound and unbound species with a characteristic curvature toward higher molecular weight as the protein concentration increases (as seen here). These bands being minor have little to no effect on the quantification of the gel, particularly since the unbound band has almost completely disappeared by this point.

For these reasons, it is unlikely that the band marked “double” is in fact the 2:1 complex, but rather a higher-order aggregate or even nonspecific binding (since it hardly enters the gel). Given this, the “single” band may also not be the 1:1 complex. The anomalous shift in the unbound band in turn introduces uncertainty in the quantification in Figure 2c, as it is no longer clear what the disappearance of the unbound band is tracking.

Again, we do not think the double band is running at an unexpected position given its mass and charge and the current gel constitution. The possibility of this band being nonspecific binding seems unlikely, the upper band appears in the gel at protein concentrations as low as single digit nanomolar and as such would represent a high affinity interaction. For us it would be difficult to envisage how a non-specific binding event could be possible at such low protein concentrations the classical view of specific and non-specific interactions between transcription factors and DNA (as we understand it). We do agree that the proximity to the top of the gel makes it difficult to distinguish between aggregates or higher order complexes. We included in our original review the data on the DNA binding domain alone which shows a similar dual banding pattern to help support our assignments.

We note that the same brachyury construct, buffer conditions and concentration ranges are present in the analysis of the single stranded EMSA shown in Figure S5a, if the full length brachyury protein were forming aggregates in the EMSA buffer then presumably would show in this gel also but that is not the case (at least not until the highest concentrations where there is possibly material close to the wells). This again is evidence to support the upper band being a DNA protein complex of relatively low mobility.

We have used a relatively simplistic quantification with the unbound DNA being the material that migrates at the position of the free probe and any shifted bands being bound. We agree that this may not be appropriate for a detailed study of the cooperatively, but this is not the purpose of our study.

In summary, a better controlled experiment is warranted if gel data were to be used for quantification for full-length brachyury. From visual inspection of Figure S5b, it is already clear that the mutant DBD (G177D) binds more strongly and with higher positive cooperativity than WT (c.f., Figure 2c). The cooperativity comparison is contrary to full-length brachyury based on the present band assignment. (Since the DNA concentration is close to the KD, there is potentially a second confounding factor from titrant depletion if total concentration is taken as the free concentration in the fit. This could directly affect the apparent Hill coefficient.) The reversal, if real, may be

interesting (and certainly relevant in view of the discussion in the text), but not supported by the electrophoretic behavior in Figure 2b.

We can see how a visual inspection of figure S5b could lead to the conclusion that there may be a difference between the shorter DNA binding domains of the WT and G177D. The gels in question were part of a series (ran in triplicate) that showed significant variability. To show this we have included below an example of a triplicate of the G177D and WT proteins. These gels are identical titrations and ran using the same apparatus yet show a large degree of internal variability (particularly for the G177D protein). When quantified we found very large error bars were too large to draw any useful conclusions as to any differences between WT and Variant. As we had better quality data for the full-length protein and it being more biologically relevant we decided to focus our quantification on the full-length proteins.

Triplicate EMSA gels of G177D (top) and WT (bottom) brachyury DNA binding domain, the protein is diluted as shown for figure s5b. The internal variability of the gels particularly the G177D (top) was too great for accurate quantification and comparison between the two variants.

We agree with the Reviewers second point about the KD of the interaction being close to the probe concentration used in the assay (1 nM radiolabelled DNA). This is somewhat a limitation of the EMSA technique and would also affect other binding assays where the readout is from binding DNA (for example FP). As we stated previously we are aware of limitations of the EMSA assay and did include a completely orthogonal assessment of binding by SPR that broadly supports our insights gained from EMSA analysis.

In summary, we believe our conclusions on the DNA binding properties of brachyury are well supported by our EMSA and SPR results. We do agree with some of the reviewer's points and are happy to make the following changes to the results section of the manuscript.

"We assume these two bands represent singly and doubly bound species; this assumption is consistent with the expected mobilities and mass to charge ratios of the complexes, and experiments..."

"We note that the DNA probe in our experiments is present at a similar concentration range to the apparent dissociation constants, giving the possibility for ligand depletion effects to affect the apparent dissociation constants, although this affect is equal across WT and G177D variants.

We have added this to the figure legend for Figure s5b

"Data on the DNA binding domain constructs was too variable for accurate quantification."

Reviewer #3 (Remarks to the Author):

Thanks for answering my questions.

Regarding Figure 3, if no further evidence is provided, at least I suggest specifying in the subtitle in the main text that this is limited to expression in the cell lines 'The Chordoma associated variant is transcribed equally to WT in cell lines'. Also, I would make it less broad and instead of 'We aimed to understand the consequence of the G177D variant within its endogenous context as other reasons for its association with chordoma could lie in protein stability or expression levels' saying something like: 'We aimed to assess if the WT and G177D variant differ in protein stability or expression levels in cell lines'.

We thank the reviewer once again for their careful review, we are happy to make these changes to the subtitle and text.

Typo in 'temperature' in Results section.

We have corrected this

Add that also CH22 together with the UmChor show equal gene expression in the Results section.

We have added this

Response to reviewers' comments 3

Reviewer #2 (Remarks to the Author):

I appreciate the authors' extensive consideration of their gel mobility shift data. The assignment of slow-moving complexes of full-length brachyury in Figure 2b and attendant conclusions about cooperative binding. If no further evidence is provided, a conservative choice may be to withhold definitive judgement on cooperativity. This does not seem a major loss given the minor difference in overall affinity of the wildtype and G177D brachyury dimer for the palindromic repeat element.

We thank the reviewer for their consideration of our response and helpful suggestions as to how to proceed. We are happy to comply with the reviewer's suggestion and modify our comments around the cooperativity of the binding. We have made the following changes to the manuscript.

The title of the results section now reads

"The G177D variant does not substantially change DNA binding affinity"

And later on, within this paragraph

"We do not see any significant differences in overall binding affinity when comparing the WT and variant proteins, with the dissociation constants [1.4 nM WT (95%CI 1.32 to 1.53) and 1.2 nM G177D (95%CI 1.03 to 1.35)] from the quantification of the data being not significantly different if using a 95% confidence interval (Figure 2c). On the other hand, differences in the degree of cooperativity are less certain. The upper and lower shifted bands on our gel appear to follow a broadly similar concentration response pattern and the hill slopes from the overall quantification [2.3 WT (95%CI 1.93 to 2.74) and 1.8 G177D (95%CI 1.46 to 2.35)] are similar. We note that these bands were not quantified independently due to uncertainties over the exact composition. Also the DNA probe in our experiments is present at a similar concentration range to the apparent dissociation constants, giving the possibility for ligand depletion effects to affect the apparent dissociation constants, although this affect is equal across WT and G177D variants."